# Transcriptional reprogramming of skeletal muscle stem cells by the niche environment

Felicia Lazure[1,2,8], Rick Farouni ®[1,3,8], Korin Sahinyan[1,2], Darren M. Blackburn[1,2], Aldo Hernández-Corchado[1,3], Gabrielle Perron ®[1,3], Tianyuan Lu ®[2,4], Adrien Osakwe ®[4], Jiannis Ragoussis ®[1,3], Colin Crist[1,2], Theodore J. Perkins ®[5,6], Arezu Jahani-Asl ®[7], Hamed S. Najafabadi ®[1,3,4] ✉ & Vahab D. Soleimani ®[1,2] ✉

Adult stem cells are indispensable for tissue regeneration, but their function declines with age. The niche environment in which the stem cells reside plays a critical role in their function. However, quantification of the niche effect on stem cell function is lacking. Using muscle stem cells (MuSC) as a model, we show that aging leads to a significant transcriptomic shift in their subpopulations accompanied by locus-specific gain and loss of chromatin accessibility and DNA methylation. By combining in vivo MuSC transplantation and computational methods, we show that the expression of approximately half of all age-altered genes in MuSCs from aged male mice can be restored by exposure to a young niche environment. While there is a correlation between gene reversibility and epigenetic alterations, restoration of gene expression occurs primarily at the level of transcription. The stem cell niche environment therefore represents an important therapeutic target to enhance tissue regeneration in aging.

Aging invariably causes a decline in muscle mass and strength. In a significant portion of the population, this leads to the onset of sarcopenia, an accelerated form of muscle wasting[1,2]. This age-related decline in muscle function is in part due to the reduction in numbers and regenerative potential of Muscle Stem Cells (MuSCs)[3]. The MuSC pool is maintained in a quiescent state under homeostatic conditions and retains the ability to activate in response to injury to repair damaged muscle tissue. The processes of activation, differentiation and self-renewal are highly dependent on niche-derived cues[4,5]. Numerous non-myogenic cells such as fibro-adipogenic progenitors (FAPs), macrophages, and endothelial cells, amongst others, have been shown to be important in regulating MuSC function[6–9]. However, the crosstalk between MuSCs and neighboring cells is perturbed in aging,

leading to a breakdown in MuSC support from the niche[10–12]. In addition to neighboring cells, age-related changes in the niche also extend to systemic factors originating from blood circulation as well as structural components such as myofibers and the composition of the extracellular matrix composition (ECM)[13–16]. As an additional layer of complexity, the niche environment is also highly dynamic and changes during diseases and throughout normal physiological aging[12,17,18]. Given the changes known to occur in the aging niche, efforts have been made to restore MuSC function by altering various components of the niche. Historically, parabiosis experiments have shown that exposure of aged muscle to young circulatory factors can restore the regenerative capacity of aged muscle[19], demonstrating the effect that extrinsic factors can have on MuSC function.

[1]Department of Human Genetics, McGill University, 3640 rue University, Montréal, QC H3A 0C7, Canada. [2]Lady Davis Institute for Medical Research, Jewish General Hospital, 3755 Chemin de la Côte-Sainte-Catherine, Montréal, QC H3T 1E2, Canada. [3]McGill Genome Centre, Victor Phillip Dahdaleh Institute of Genomic Medicine, 740 Dr Penfield Avenue, Montreal, QC H3A 0G1, Canada. [4]Quantitative Life Sciences, McGill University, Montreal, Canada. [5]Sprott Center for Stem Cell Research, Ottawa Hospital Research Institute, 501 Smyth Road, Ottawa, ON K1H 8L6, Canada. [6]Department of Biochemistry, Microbiology and Immunology, University of Ottawa, Ottawa, ON K1H 8M5, Canada. [7]Department of Cellular and Molecular Medicine and University of Ottawa Brain and Mind Research Institute, University of Ottawa, 451 Smyth Road, Ottawa, ON K1H 8M5, Canada. [8]These authors contributed equally: Felicia Lazure, Rick Farouni. ✉e-mail: hamed.najafabadi@mcgill.ca; vahab.soleimani@mcgill.ca

Recent transplantation studies have further demonstrated the influence of the niche environment. In one study, it has been shown that MuSCs can be partially transcriptionally reprogrammed to suit their location of engraftment when injected into a site that is not their niche of origin[20]. In a human muscle xenograft model, it was shown that regenerative potential was maintained when transplanting both aged and young human muscle grafts into young immune-deficient mice[21], once again placing importance on the host environment. While there is clear evidence for reversibility of the aging muscle phenotype, other studies provide evidence supporting the presence of irreversible cell-intrinsic changes in aged MuSCs. For example, it has been reported that aged MuSCs have a reduced capacity to engraft into healthy adult mice compared to young counterparts[22,23]. Despite the evidence for both cell-intrinsic and niche-mediated changes occurring in aged MuSCs, models to effectively decouple the two have been lacking. Studying MuSCs within the context of their niche and without ex vivo expansion has previously been challenging due to the scarcity of cellular material sufficient for genome-wide analyses.

Here, we report the use of allogeneic stem cell transplantation combined with Switching Mechanism at 5′ End of RNA Template (SMART-Seq) technology to directly quantify the effect of the niche environment on the MuSC gene expression profile. We examined the genome-wide reversibility of age-related altered gene expression by exposure of MuSCs to the young niche environment. Combined with single-cell RNA-seq, ATAC-seq, and Whole Genome Bisulfite (WGBS) sequencing data, we characterized the transcriptomic and epigenetic profile of genes that are reversible and irreversible by the exposure of aged MuSCs to young niche environment.

## Results

### MuSC subpopulations display different dynamics of loss versus retention in aging

To determine the effect of aging on the population dynamics of MuSCs and muscle-resident niche cells, we performed single-cell RNA sequencing of MuSCs, as well as two important non-myogenic niche cell populations: macrophages[24,25] and Fibro/Adipogenic progenitor cells (FAPs)[11] from young (5–6 weeks old) and aged (24 months old) mice (Fig. 1a, Supplementary Fig. 1, Supplementary Data 1). For the purpose of quality control, we assessed the correlation between our scRNA-Seq data from MuSCs and the published Tabula Muris dataset[26] (Supplementary Fig. 2). To selectively increase the number of MuSCs, FAPs and macrophages for clustering and downstream analysis, we used FACS to enrich for the three cell types mentioned as an alternative to sequencing whole muscle cell suspension (Fig. 1b, Supplementary Fig. 1). In addition to an overall reduction in numbers (Fig. 1b–e, Supplementary Fig. 1), UMAP clustering of scRNA-seq data suggest that MuSCs exhibit an age-related shift in their population diversity (Fig. 1e, f). On the other hand, when comparing young and aged FAPs, the primary effect of aging is an overall expansion in cell number in aged animals (Fig. 1d–g). Additionally, while FAPs appear to be major contributors to the components of the Extracellular Matrix (ECM), FAPs from aged mice exhibit reduced expression of ECM genes such as *Col1a1* and *Col1a2* among others (Fig. 1h, k, Supplementary Fig. 3a–e, Supplementary Data 1). The regulation of ECM stiffness and composition is key to maintain proper MuSC function[14,27], therefore this deregulation in matrisome genes between young and aged FAPs may be a direct contributor to ECM deregulation. Macrophages, in contrast, do not show a striking change in numbers nor a major shift in population dynamics during aging (Fig. 1i). However, a dysregulation in the expression of previously identified macrophage marker genes such as *Msrb1*[28], *Plcb1*[29], and *Gsr*[30] can be observed, possibly indicating a phenotypic switch during aging (Fig. 1j, l, Supplementary Data 1).

Evidence suggests that, during aging, cells are lost due to the accumulation of DNA damage and oxidative and replication stress that occur over time[31,32]. Our data suggest that an alternative mechanism

may also be at play since different subpopulations of MuSCs are dissimilarly lost or retained in aging (Fig. 1e). For example, rather than a stochastic loss, our scRNA-seq data shows that Cluster 1 (MuSC1) is more prone to age-related loss whereas the cells in Cluster 2 (MuSC2) display enhanced retention with age (Fig. 1e). Importantly, this age-related shift in populations is more prominent in MuSCs compared to FAPs and macrophages, which display a more stochastic change in populations during aging (Fig. 1e, g, i). Taking a closer look at MuSC population dynamics, MuSCs from young mice segregate into three major clusters based on UMAP clustering (Fig. 2a, b). While all cell clusters express typical MuSC marker genes, such as Pax7, Myf5, ITGA7, and VCAM1, cell clusters differ with regards to quiescence markers and more (Supplementary Fig. 1h). MuSCs belonging to Cluster 3 (MuSC3) are characterized by genes involved in MuSC activation and differentiation such as *MyoD1*, *Cdkn1c*, *Megf10*[33] and *Myog*[34] (Fig. 2b-f, Supplementary Fig. 1h). Additionally, this cluster expresses reduced amounts of quiescence markers compared to other clusters, including *Notch3*[35], *Calcr*[36], *Chrdl2*[37] (Supplementary Fig. 1h). Also elevated in this cluster are the cell cycle genes *Ccnd1* and *Cdkn1a*, which correlates with the elevation of CyclinD1 and P21 proteins in young EDL-associated MuSCs (Supplementary Fig. 4). The heterogeneity of *Ccnd1* expression between scRNA-Seq clusters is also corroborated by immunofluorescence staining of varying intensity in young MuSCs (Supplementary Fig. 4a–c) at 0 h post isolation. Importantly, MuSC Cluster 3 is completely lost in aged mice (Fig. 3a,b), likely due to reduced propensity for early activation and delayed entry into cell cycle of aged MuSCs compared to young[38] (Supplementary Fig. 4).

While MuSC clusters 1 and 2 are very similar in their gene expression profiles, there are certain genes and pathways that differentiate the two (Fig. 2c, e, f). MuSC Cluster 1 is also reduced in aging, albeit to a lesser extent than Cluster 3 (Fig. 1e). MuSC Cluster 1 is characterized by increased activation of the Fos/Jun/Egr3 pathway, which has been shown to be a signature of the most pro-regenerative MuSCs[39] (Fig. 2f). In contrast, MuSC Cluster 2 which is retained and augmented in aged mice displays increased expression of stress response and antioxidant genes[40,41], such as *Mt1, Mt2 Txnrd1* and *Gpx3* (Supplementary Fig. 5). For example, Gpx3 has been shown to be required for the survival of human MuSCs under stressful conditions, by reducing cytotoxic damage caused by oxidative stress[42]. The activation of the stress response pathway by MuSC2 may provide these cells with the resilience to cope with increased oxidative stress and genotoxicity of the aging niche (Supplementary Fig. 5). Additionally, MuSC Cluster 2 is demarked by an elevated expression of OSMR (Fig. 2f); OSMR activation can induce the JAK/STAT pathway, which is known to be over-activated in aging[43].

Since both MuSCs and FAPs display significant heterogeneity as evidenced by the presence of distinct clusters, we performed cluster-specific gene expression analysis between young and aged cells. Notably, we observe a strong correlation between age-affected genes on both of the main clusters of MuSCs, implying a consistent transcriptomic shift in different MuSC clusters that persist through aging (Fig. 3c, d). A similar effect is observed in FAPs (Fig. 3d), which are present within the same niche. This data suggests that, despite differences in subpopulation loss, aging impinges upon common gene networks and pathways, thus displaying a broad effect on MUSCs. To detect and compare the cell-cell interactions occurring between MuSCs, FAPs and macrophages during aging, we used the scDiffCom analysis[44]. We grouped Emitter (ligand source) and Receiver (receptor source) cells by cell type (FAP, MuSC and Macrophage). We grouped cell-cell interactions under 4 categories: UP-regulated in aging (log-fold changes>log2 (1.5), p-values < 0.05), DOWN-regulated in aging (log-fold changes < −log2 (1.5), adjusted p-values < 0.05), FLAT (absolute log-fold changes<log2 (1.5), adjusted p-values > 0.05) and NSC or non-significant (absolute log-fold changes>log2 (1.5), adjusted p-values > 0.05). This analysis revealed that receiver/emitter

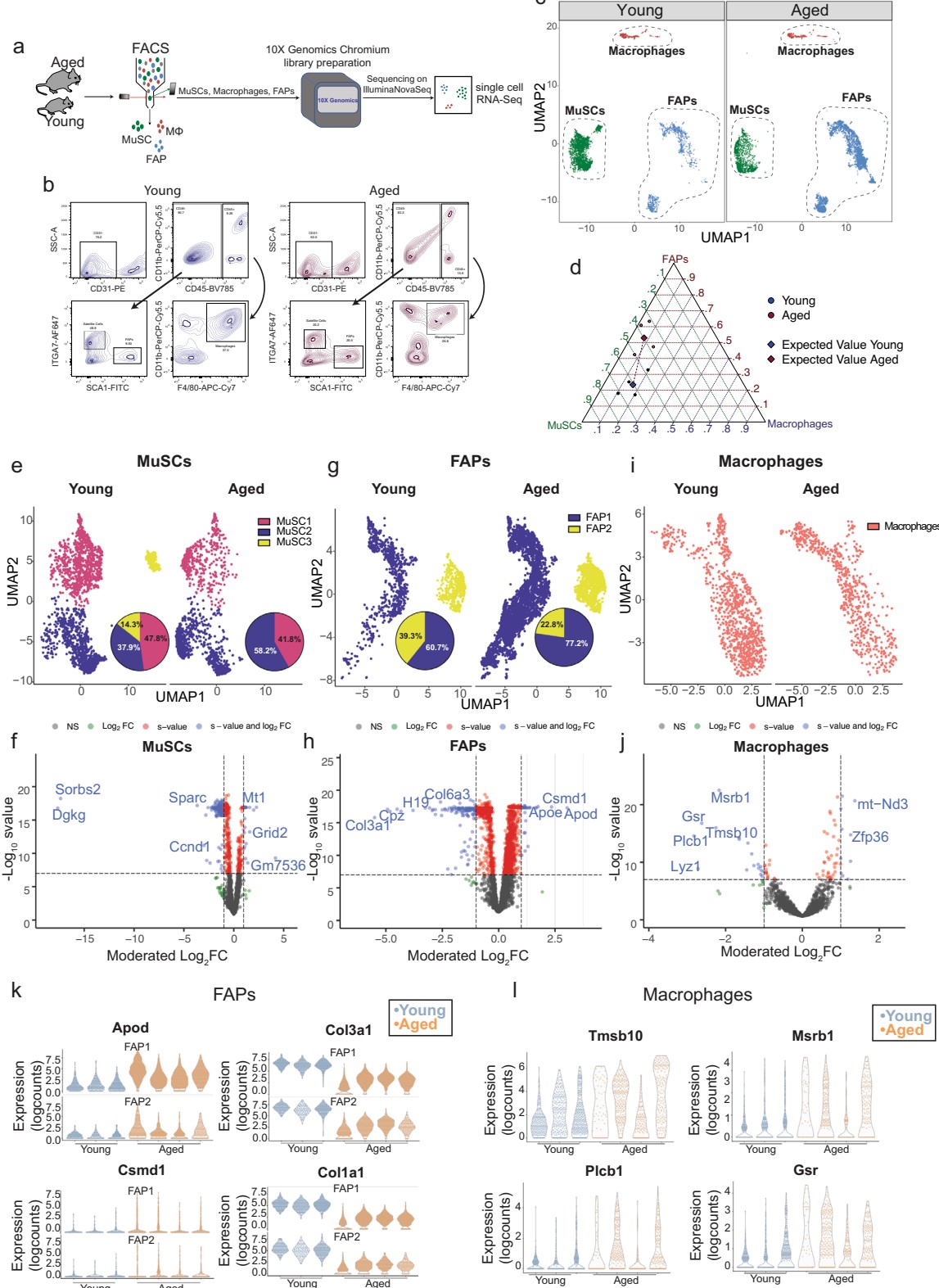

interactions involving Extracellular Matrix (ECM) go down in aging while interactions involving chemokine signaling go up during aging (Supplementary Fig. 6a). Next, over-representation scores of the Emitters, Receivers, and Emitter–Receiver pairs showed that FAPs are over-represented as emitters and receivers in young. Moreover, FAP-FAP, MuSC-FAP, and FAP-MuSC cell interactions are over-represented in young mice. On the other hand, The MuSC- MuSC and Macrophage–Macrophage interactions are over-represented in

aged samples (Supplementary Fig. 6b). Together this data suggests that communications between MuSCs and non-myogenic muscle-resident cells are altered during aging.

## Heterochronic transplantation into a young niche can transcriptionally reprogram aged MuSCs

Given that aging affects common gene networks and pathways regardless of the subpopulation (Fig. 3c, d) in both MuSCs and FAPs,

**Fig. 1 | The population dynamics of MuSCs and niche cells are altered in aging. a** Schematic diagram of the workflow for the isolation and single-cell RNA sequencing of young and aged MuSCs, macrophages and Fibro-Adipogenic Progenitors (FAPs).**b** Fluorescence-Activated Cell Sorting (FACS) plots demonstrating the strategy for the simultaneous isolation of MuSCs, macrophages and FAPs from young (5–6-week-old) and aged (24-month-old) mice.**c**, Global UMAP plot of scRNA-seq data from young and aged MuSCs, FAPs, and macrophages ($n = 3$ mice per group) **d** Dirichlet Regression analysis of the proportions of sequenced MuSCs, FAPs and macrophages from young and aged mice. The best-fitting model was selected based on the likelihood-ratio-test ($p$-value $< 2.2e^{-16}$; See notebook at https://csglab.github.io/transcriptional_reprogramming_muscle_cells/). **e** UMAP plot of scRNA-seq data from young and aged MuSCs including pie charts showing the proportions of cells in each subpopulation. **f** Volcano plots showing genes that are up or downregulated due to aging in MuSCs at a threshold of moderated LFC > 1 and $s$-value < 0.05. **g** UMAP plot of scRNA-seq data from young and aged FAPs including pie charts showing the proportions of cells in each subpopulation. **h** Volcano plots showing genes that are up or downregulated due to aging in FAPs at a threshold of moderated LFC > 1 and $s$-value < 0.05. **i** UMAP plot of scRNA-seq data from young and aged macrophages. **j** Volcano plots showing genes that are up or downregulated due to aging in macrophages at a threshold of moderated LFC > 1 and $s$-value < 0.05. **k** Violin plots of select representative genes that are differentially expressed between young and aged FAPs. **l** Violin plots of select representative genes that are differentially expressed between young and aged macrophages (moderated LFC > 1, $s$-value < 0.05).

---

we reasoned that the alterations in the MuSC transcriptome may result from changes in their niche environment during aging. We therefore decoupled age-related alterations in the MuSC transcriptome that are cell-intrinsic from changes caused by the aging niche. To this end, we developed an in vivo model to quantify the effect of the niche environment on the MuSC transcriptome (Fig. 4a). We first used FACS to isolate Pax7-nGFP MuSCs from the hindlimbs of mice aged 22-26 months[45,46] (Fig. 4a), followed by immediate injection of 10,000-20,000 of these freshly sorted aged donor MuSCs into the irradiated, host MuSC-depleted hindlimb of living young NOD-*Prkdc*[em26Cd52]*Il2r-g*[em26Cd22]/NjuCrl (NCG) mice (Fig. 4a–e, Materials and Methods). The age of the donor cells had no significant effect on engraftment efficiency (Supplementary Fig. 7). Aged donor MuSCs were left to home to the young host niche for three weeks (Fig. 4e). Importantly, donor MuSCs return to quiescence during this time, as shown by the expression of Pax7 in donor GFP⁺ donor cells, and the lack of ki67 expression compared to injured controls (Fig. 4e, Supplementary Fig. 8). After 21 days of residing in the young niche, the engrafted Pax7-nGFP MuSCs were reisolated by FACS (Fig. 4f, Materials and Methods) for RNA-seq library preparation, to allow a direct transcriptome comparison of MuSCs from the same donor mouse before and after transplantation.

To control for the effect of engraftment itself, MuSCs from young mice (5–6 weeks old) were also injected into host mice of similar age (Fig. 3g, Materials and Methods) (Fig. 4g). Since MuSC recovery yielded only 30–400 cells on average, we used SMART-seq technology to create sequencing-ready libraries of cells before ($T_0$) and after engraftment ($T_{21}$), as it allows the reverse transcription of minute starting quantities of RNA[47] (Supplementary Fig. 9, Materials and Methods). Genes altered by aging in our SMART-seq bulk libraries showed high concordance with age-affected genes from our independent scRNA-seq dataset (Supplementary Fig. 9g). After sequencing, age-altered genes were classified into those showing an effect of age (different between young $T_0$ and aged $T_0$), an effect of engraftment (control) (different between young $T_0$ and young $T_{21}$), and an effect of exposure to the heterochronic niche, defined as the effect of heterochronic transplantation while accounting for the effect of engraftment ($\Delta niche = \Delta_{AgedT21-AgedT0} - \Delta_{YoungT21-YoungT0}$) (Fig. 4g–i, Supplementary Table 1, Supplementary Data 2, 3, supplementary Fig. 10). For example, age-affected genes that remain unchanged by the heterochronic niche include *B2m, Igf1, Cdkn3, Mt1*, and *Birc5*, amongst others, indicating the presence of niche-irresponsive age-related defects (Fig. 5a-c, Supplementary Data 2,3).

To identify genes in aged engrafted MuSCs whose expression was reversible by engraftment into young hosts, we focused on those genes that showed an effect of both age and of niche in their expression profile (Fig. 5). We designated these age-related genes whose expression is reversible as Age-Related Reversible (ARR), and genes whose expression is irreversible as Age-Related Irreversible (ARI). It is important to note that, due to the nature of our transplantation model which includes age-related variability as well as variation in engraftment efficiency, not all ARRs are reproducible. Therefore, we set various threshold cutoffs to demarcate significant ARRs. At a stringent

cutoff of $s$-value < 0.05 and moderated LFC > 1, out of 397 genes with altered expression in aging, 185 genes (46.6%) were transcriptionally reprogrammable by exposure to the young niche (Fig. 5e, Supplementary Data 2). Similarly, at a more permissive threshold of $s$-value < 0.15, moderated LFC > 1 and batch LFC > 0.5, 686 ARRs were identified (53.6%) out of 1280 genes that are differentially expressed in aging (Fig. 5e, Supplementary Data 3).

Notable among these ARRs are genes whose proteins contribute to ECM composition and remodeling such as *Col3a1, Col4a1, Col4a2, Mmp2*, and *Sparc*[48], genes involved in cell cycle regulation such as *Cdc20 and Eya2*, as well as the cytokine aging factor *Ccl11*[49] (Fig. 5c, Supplementary Fig. 11, Supplementary Data 2, 3). The restorative ability of the niche is illustrated when plotting genes whose expression is affected by age and by their response to heterochronic niche, showing a clear negative correlation (Fig. 5d). Interestingly, some markers of the previously characterized MuSC Cluster 3 are among the panel of downregulated ARRs, which may be indicative of a partial restoration of this cluster post-transplantation (Supplementary Fig. 12, Supplementary Data 3). Overall, this data shows that the niche is a principal regulator of the MuSC transcriptome. More importantly, a significant portion of the age-related altered transcriptome of MuSCs can be reprogrammed back to the youthful state by exposure to young niche milieu.

## The epigenetic signature affects the transcriptional response of MuSCs to the niche environment

To gain mechanistic insight into the age-related alterations in the MuSCs transcriptome and the effect of the niche environment on their transcriptional reprogramming, we first asked whether post-transcriptional and/or epigenetic mechanisms, such as changes in RNA stability or the chromatin state respectively, were playing a role. We examined changes in RNA stability associated with aging and/or niche response in MuSCs using the RNA-seq data described in the previous section. We deconvolved the transcriptional and post-transcriptional effects based on the relative abundance of pre-mRNA and mature mRNA[50] (Fig. 6 a, b, Materials and Methods). This approach determined that differential RNA stability is not a major mechanism of either the age-related alterations in the transcriptome or the niche-mediated restoration of age-related genes (Fig. 6a, b).

To investigate whether the differential responsiveness to the niche observed between ARIs and ARRs was a manifestation of different chromatin states in young and aged MuSCs, we assessed the frequency of CpG islands in the vicinity of the TSS of genes with age-related changes in expression as a proxy for their capacity to be influenced by DNA methylation (Fig. 6c). We found a slight increase in the frequency of CpG islands associated with ARIs compared to ARRs (Fig. 6c) Next, we performed Whole Genome Bisulfite Sequencing on primary myoblasts derived from young and aged MuSCs (Supplementary Data 4, Materials and Methods). From the WGBS data, we observed that ARIs tend to associate with a higher number of differentially methylated regions (DMRs) within candidate *cis*-regulatory elements (cCREs) per gene (Fig. 6d). However, when we begin from the

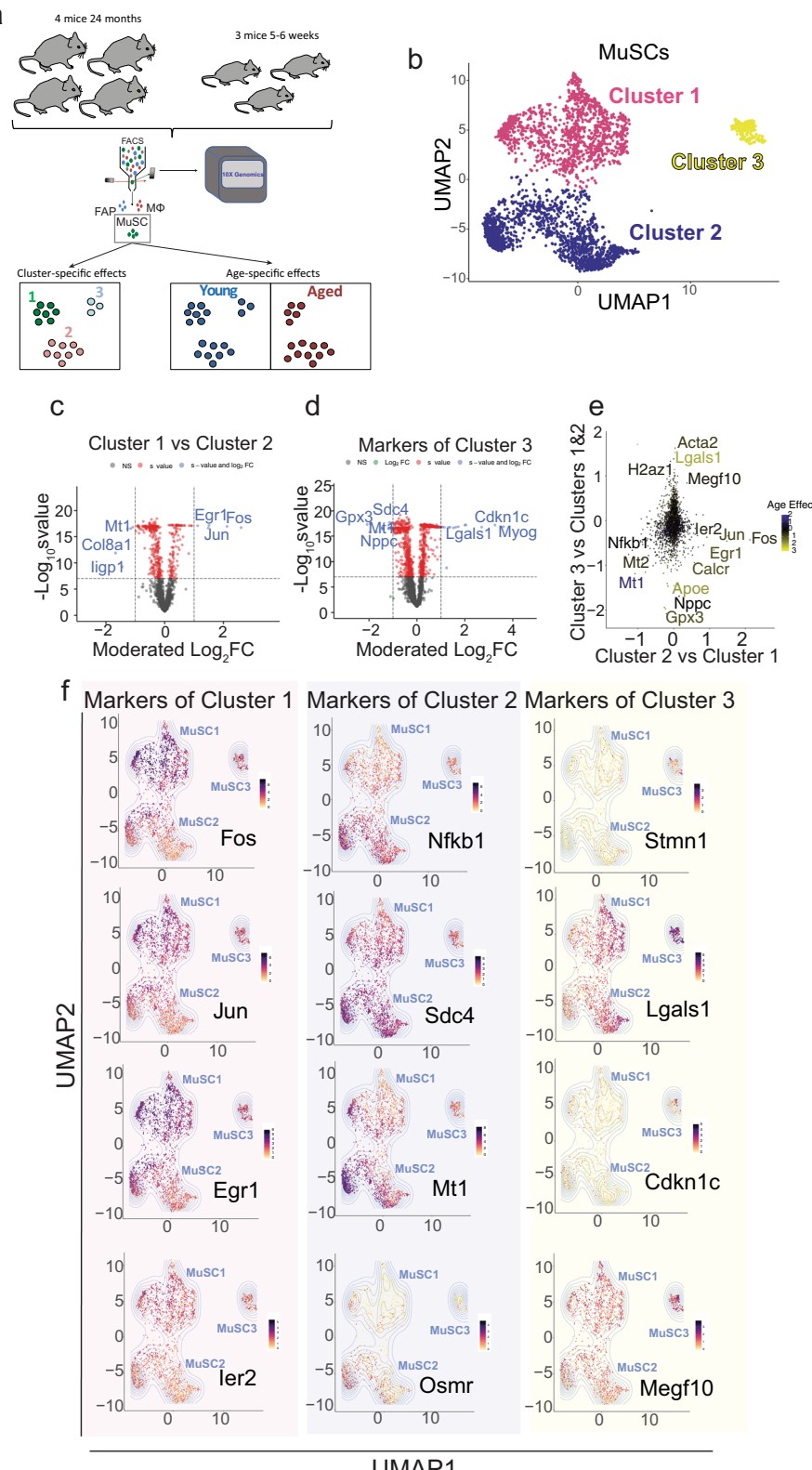

**Fig. 2 | MuSCs can be classified into three main clusters. a** Schematic diagram of the scRNA-seq analysis on young and aged MuSCs. **b** UMAP embedding of MuSCs, colored based on cluster. **c** Volcano plots showing genes that are up or down-regulated in Cluster1 compared to Cluster2 of MuSCs, at a threshold of moderated LFC > 1 and $s$-value < $10^{-8}$. **d** Volcano plots showing genes that are up or down-regulated in Cluster 3 compared to Clusters 1 and 2, at a threshold of moderated LFC > 1 and $s$-value < $10^{-8}$. **e** Scatterplot showing the moderated LFC of differentially expressed genes between Clusters MuSC2 and MuSC1, and between MuSC3 Cluster compared to Clusters MuSC1 & MuSC2. Blue and yellow represent down and upregulated genes respectively during aging. **f** Gene expression plots in the UMAP embedding for select representative positive marker genes of each cluster ranging from highest (black) to lowest (yellow) expression.

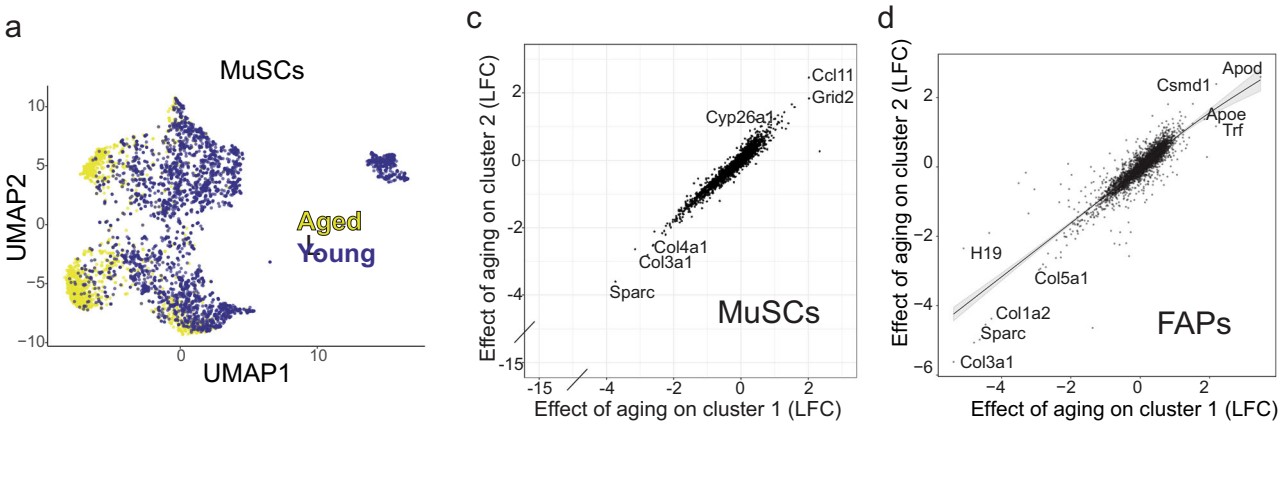

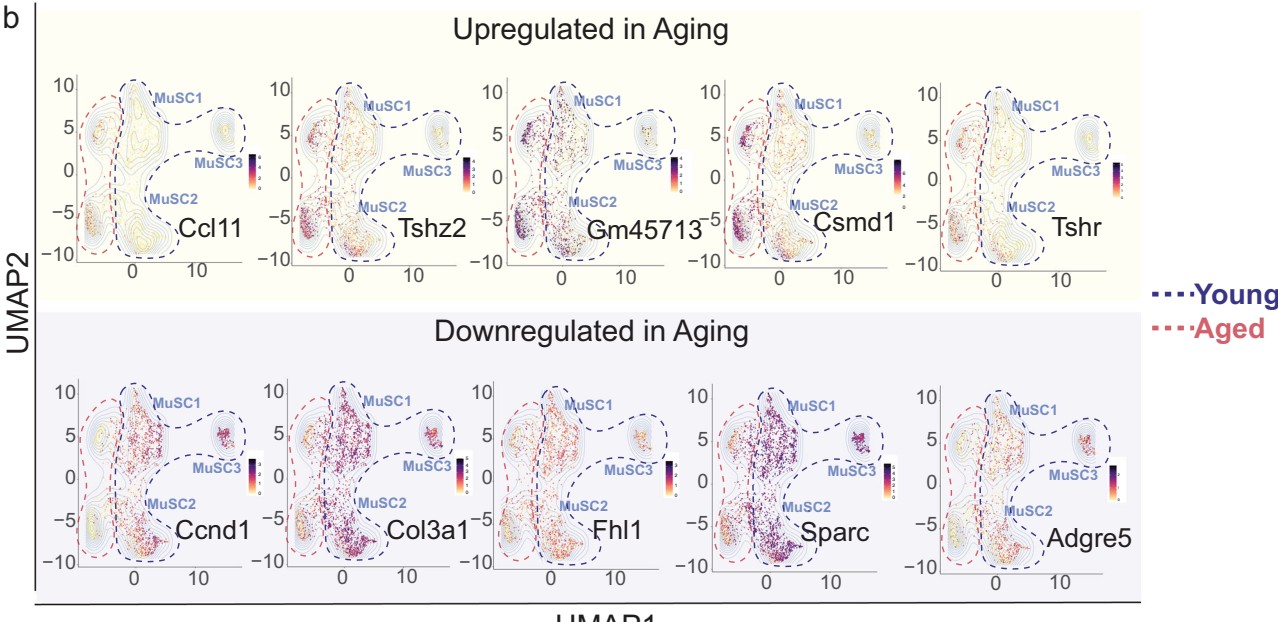

**Fig. 3 | Aging impinges upon common genes across distinct clusters of MuSCs.**
**a** UMAP embedding of MuSCs, colored based on age. **b** Gene expression plots in the
UMAP embedding for select representative genes that are up or downregulated in
aging, ranging from highest (black) to lowest (yellow) expression. **c** Scatterplot
showing the concordance of the moderated LFC effect of age on Clusters MuSC2
and MuSC1. **d** Scatterplot showing the concordance of the moderated LFC effects
of age on FAP1 and FAP2 clusters in FAPs.

lists of DMRs and classify genes that are associated with at least one
DMR, we did not find a significant over-representation of ARIs among
the list of DMRs (Supplementary Table 2).

On a global scale, we observed no net gain or loss of methylation
at DMRs, which is consistent with previous studies (Fig. 6e)[51]. Similar to
what has been shown in various cancers[52,53], most age-related DMRs
map outside of the Transcription Start Sites (TSS), suggesting a
potential role for distal enhancer elements (Fig. 6e). To establish
whether age-related changes in DNA methylation leads to changes in
chromatin accessibility, we performed Assay for Transposase-
Accessible Chromatin using sequencing (ATAC-seq) on 5000 freshly
sorted MuSCs isolated per sample from young and aged mice (Sup-
plementary Figs. 13, 14, Materials and Methods). By assessing chro-
matin accessibility spanning the DMRs, we observed that age-related
gains of methylation lead to a decrease in chromatin accessibility
(Fig. 6f), and that age-related losses in methylation are associated with
increases in accessibility (Fig. 6f-g). We identified a total of 9648 ATAC-
seq peaks with significant age-associated changes in chromatin
accessibility (LFC > 1, $p_{adj}$ < 0.05) (Supplementary Data 5), which lead

to age-accessible and inaccessible transcription factor binding sites
(Supplementary Fig. 15). As expected, we observed significant overlap
between upregulated genes and those associated with age-accessible
cREs, as well as downregulated genes and those associated with age-
inaccessible cREs (Fig. 6h). While ARI genes were not significantly
more likely to be associated with a differentially accessible peak
compared to ARRs (Fig. 6h, Supplementary Data 2, Supplementary
Table 3), there is a slight increase in the percentage of upregulated
ARIs associated with differentially accessible peaks compared to
upregulated ARRs (Fig. 6i). Furthermore, when we examine the aver-
age number of differentially accessible peaks per gene at various dis-
tances from the TSS, we observe that, as we move closer to the TSS, a
pattern emerges whereby upregulated ARIs are associated with a
higher average number of age-accessible ATAC-seq peaks (Fig. 6j).
Together, these data suggest that distinct epigenetic changes may
influence the responsiveness to niche-induced transcriptional repro-
gramming (Fig. 6c–j). However, further research is required to provide
a causal relationship between chromatin changes and gene respon-
siveness to niche-derived cues.

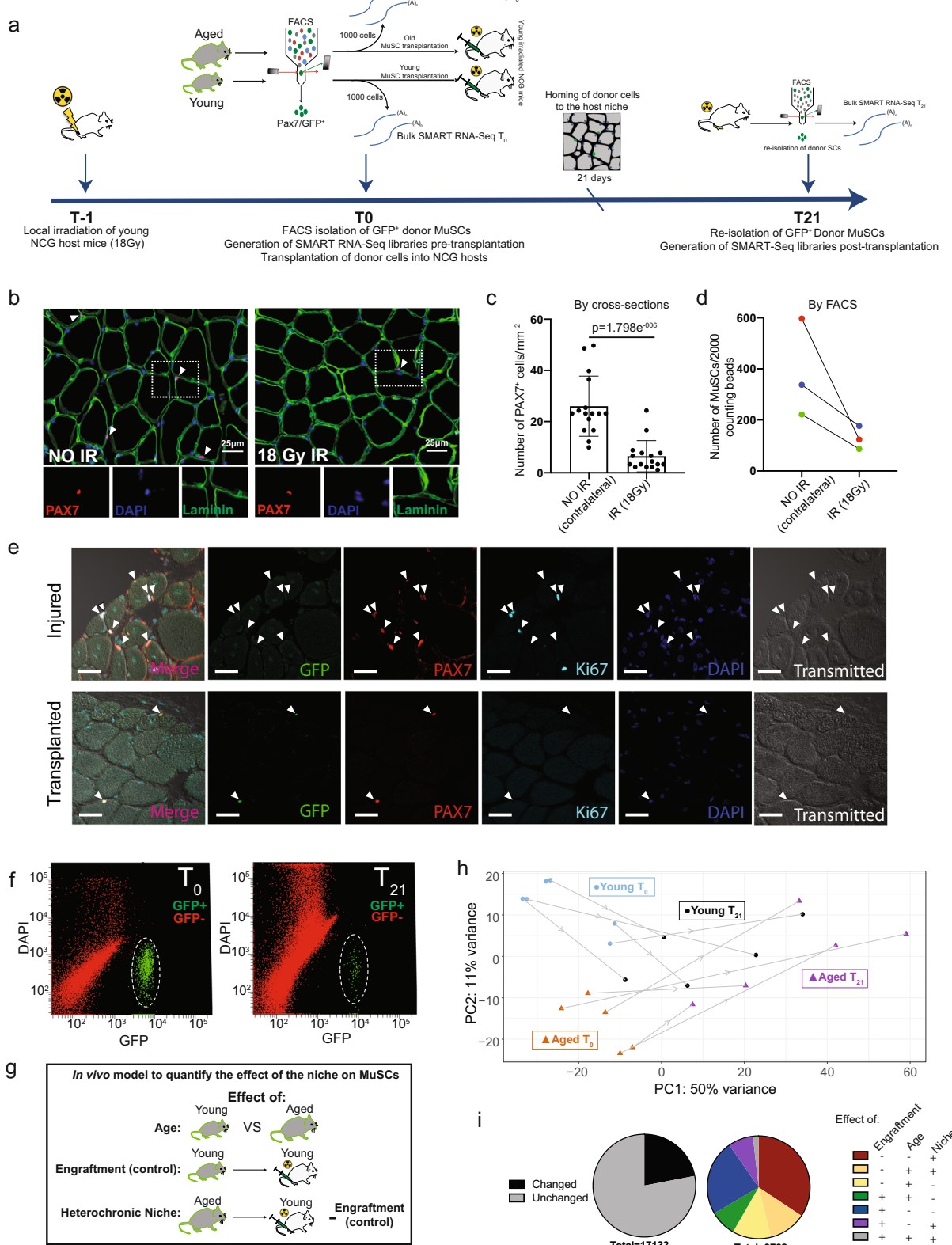

## Discussion

The extent to which the age-related changes in MuSC function are a cause or consequence of aging muscle tissue has important implications for our understanding of skeletal muscle regeneration in healthy and aging skeletal muscle. Here, we have decoupled age-related cell-intrinsic effects, niche-mediated cell-extrinsic effects, and changes in population dynamics of MuSCs and two key muscle-resident cells in

young and aged mice. First, we showed that the age-related reduction in MuSCs is not stochastic, as specific subpopulation gene expression signatures are preferentially lost or retained in aged mice (Figs. 1, 2). The differential loss/retention of subpopulations of MuSCs such as MuSC Cluster 2 may be in part due to their ability to upregulate stress response genes, protecting them from a genotoxic environment during aging (Supplementary Fig. 5). Due to the complete loss of the

**Fig. 4 | Development of an allogeneic stem cell transplantation model to directly quantify the effect of the niche environment on the transcriptome of MuSCs. a** Schematic diagram of the pipeline for allogeneic MuSC transplantation. **b** Immunofluorescent staining for PAX7, Laminin and DAPI on cross-sections from TA muscles that have been treated with 18 Gy of irradiation compared to the contralateral untreated TA muscle ($n = 2$ mice) **c** Quantification of the number of Pax7$^+$ MuSCs/mm$^2$ in irradiated compared to untreated contralateral Tibialis Anterior (TA) muscle cross-sections ($n = 2$ mice, $n = 16$ cross-sections counted per group, two-tailed $t$ test). Data are presented as mean values SD. $p = 1.798e^{-006}$. **d** Quantification of the number of ITGA7$^+$/Lin$^-$ MuSCs per 2000 FACS counting beads sorted from irradiated compared to untreated contralateral TA muscles ($n = 3$ mice) **e** Immunofluorescent staining for PAX7, GFP, KI67 and DAPI of TA muscles from host mice 21 days post-transplantation with donor cells and cardiotoxin-injured TA muscles, showing engraftment and homing to the niche of GFP$^+$ donor MuSCs, and lack of KI67 in engrafted TAs compared to injured controls. Scale bar = 25 μm (3 independent repeats were performed) **f** Representative FACS plots showing the isolation of GFP$^+$ MuSCs at T$_0$ (left), followed by the re-isolation of the same cells after engraftment into young NCG mice at T$_{21}$ (right). **g** Schematic diagram of the separation of age, engraftment, and niche effects after allogeneic MuSC transplantation. **h** Principal Component Analysis (PCA) plots of bulk RNA-seq data, from young and aged mice, pre- (T$_0$) and post-transplantation (T$_{21}$). **i** Pie chart of the relative proportion of genes that are altered after transplantation into the host niche versus the genes that remain unchanged after transplantation (left). Pie chart of the relative proportions of genes showing an effect of engraftment, age, or the allogeneic niche, in all combinations, from transplantation RNA-seq datasets (right). (Moderated LFC > 1, s-value < 0.05).

committed MuSC Cluster 3 and partial loss of the pro-regenerative MuSC Cluster 1, one possibility is that the cells that have a competitive advantage to respond to the aged niche environment are retained at the expense of the cells that are maintaining tissue health.

Second, we demonstrate that, despite differences in the transcriptomes of the MuSC clusters, the effect of age on gene expression is largely uniform (Fig. 3 c). The uniform effect of age on gene expression is present in both MuSCs and FAPs, suggesting that the niche environment has a fundamental role in the age-related changes in MuSC gene expression (Fig. 3c, d).

Third, while it is known that aging alters the composition of the MuSC niche, the direct effect of the niche on gene expression in the residing stem cells has remained unexplored. Using an allogeneic stem cell transplantation model (Fig. 4), we showed that a significant fraction of the changes in the transcriptome of aging MuSCs can be reversed by exposure to the young muscle environment and are therefore niche-responsive (ARRs) (Fig. 5). Equally important, our allogeneic transplantation model allowed us to also tease out a panel of genes whose expression is irreversible, or niche irresponsive (ARIs) (Fig. 5). Given the high percentage of reversibility in gene expression, our findings indicate that age-related changes in the niche are principal drivers of the resulting alterations in the MuSC transcriptome (Fig. 5).

Fourth, we found that aging is correlated with changes at the level of chromatin accessibility and DNA methylation in MuSCs (Fig. 6, Supplementary Figs. 12-13, Supplementary Data 4–5). We found that age-related loss of DNA methylation is associated with loss of heterochromatin. We also observe an association between age-upregulated ARIs and increased accessibility of associated ATAC-Seq peaks in aging. A possibility here is that the transcriptional response of MuSCs to the niche environment is dependent on whether their epigenetic signature has been altered, suggesting that changes in chromatin accessibility and DNA methylation may impede transcriptional restoration by niche exposure. This concept is consistent with other findings on stem cell reprogramming, where the remodeling of cells' epigenetic signature by forced expression of transcription factors is required to return cells to a more pluripotent or progenitor-like state[54,55]. In the case of age-downregulated genes, however, a different epigenetic state between ARRs and ARIs was not observed, indicating that these ARRs may be responding to more transient signaling pathways within the niche. Therefore, further research is required to identify specific signatures that confer plasticity in MuSC gene expression in response to the niche. Furthermore, 3D mapping of genome architecture, using Hi-C or a related technique, would be required to establish a causal relationship between gene reversibility and changes in chromatin state.

In summary, our studies highlight the importance of the muscle stem cell niche environment as a critical regulator of MuSC gene expression. The plasticity of the MuSC transcriptome suggests that modulating the niche environment can be a powerful tool to restore stem cell-mediated endogenous muscle regeneration in aging. Consequently, as opposed to focusing solely on MuSCs themselves to mitigate the effects of aging on MuSCs, bioengineering of the niche in its entirety may be a viable therapeutic option. Our findings also have important implications for preventative therapies, suggesting that potential treatments targeted to the niche would be most effective if administered prior to both the initiation of age-related changes in population dynamics and the onset of alterations in the chromatin state.

## Methods
### Isolation of pure populations of MuSCs and niche cells by fluorescence-activated cell sorting (FACS)
Hindlimb muscles were dissected from young (4–6 weeks old) or aged (22–26 months old) C57BL/6J (Jackson Laboratory, 000664) mice and minced until no visible tissue chunks were visible. Muscle was then digested in a 15 ml Falcon tube containing 2.4 U/ml Collagenase D (11088882001, Roche), 12 U/ml Dispase II (04942078001, Roche) and 0.5 mM CaCl$_2$ in Ham's F10 media (11550043, Gibco) for two periods of 30 min each on a shaker in a 37 °C/5.0% CO$_2$ Tissue Culture (TC) incubator. After the first 30 min, the remaining muscle chunks were pelleted by centrifugation at 600 × $g$ for 25 s and the supernatant containing released mononuclear cells was supplemented with 9 ml Fetal Bovine Serum (FBS) (080450, Wisent). The remaining pellet of undigested muscle tissue was triturated, and additional digestion buffer was added for the second 30-min incubation. The combined digested tissue was then filtered through a 40 μm cell strainer (C352340, Falcon) and spun down at 600 × $g$ for 18 min. The supernatant was discarded and the cell pellet was dissolved in 500 μl 2% FBS/PBS (v:v) with 0.5 mM EDTA, and filtered once more. For isolation of GFP + MuSCs from Pax7-nGFP mice, 0.5 μg of DAPI (Invitrogen, D3671) was added for cell sorting of the DAPI$^-$/GFP$^+$ population using a FACSAria Fusion cytometer (BD Biosciences). For simultaneous isolation of MuSCs, FAPs and macrophages, the digested tissue was supplemented with FC (fragment crystallizable) block (clone 2.4G2). Antibodies for anti-mouse ITGA7-AF647 (R&D systems, FAB3518R, 1:500), anti-mouse CD31-PE (Invitrogen 12-0311-81, 1:5000), anti-mouse Sca1-FITC (BD Pharmigen, 557405, 1:5000), anti-mouse CD45-BV785 (Biolegend, 103149, 1:5000), anti-mouse CD11b-PerCP-Cy5.5 (Biolegend, 101227, 1:5000), anti-mouse F4/80-APC-Cy7 (Biolegend, 123117, 1:1000), and Hoechst 33342 (H1399, Molecular Probes) were added and the sample was incubated at room temperature (RT) for 15 min with intermittent shaking. The sample was then washed with 5 ml of 2% FBS/PBS (v:v) with 0.5 mM EDTA and spun down at 600 × $g$ for 15 min. The stained cells were then resuspended in 700 μl of 5% FBS/PBS (v:v) with 0.5 mM EDTA and filtered through a 40 μm filter before sorting. Color compensation was done by staining 20 μl of UltraComp eCOMP beads (01-2222-41, Invitrogen) in 200 μl of 2% FBS/PBS (v:v) and calculated using DIVA before sorting on a FACSAria fusion III cytometer (BD Biosciences). In order to maintain the ratios of MuSCs, macrophages and FAPs, all cells were sorted into a single tube and 2500 cells were captured per sample for single-cell RNA-Sequencing using 10X Genomics chromium platform. FACS gating strategies benefited from Pasut et al.[56], Farup et al.[57], and Low et al.[58].

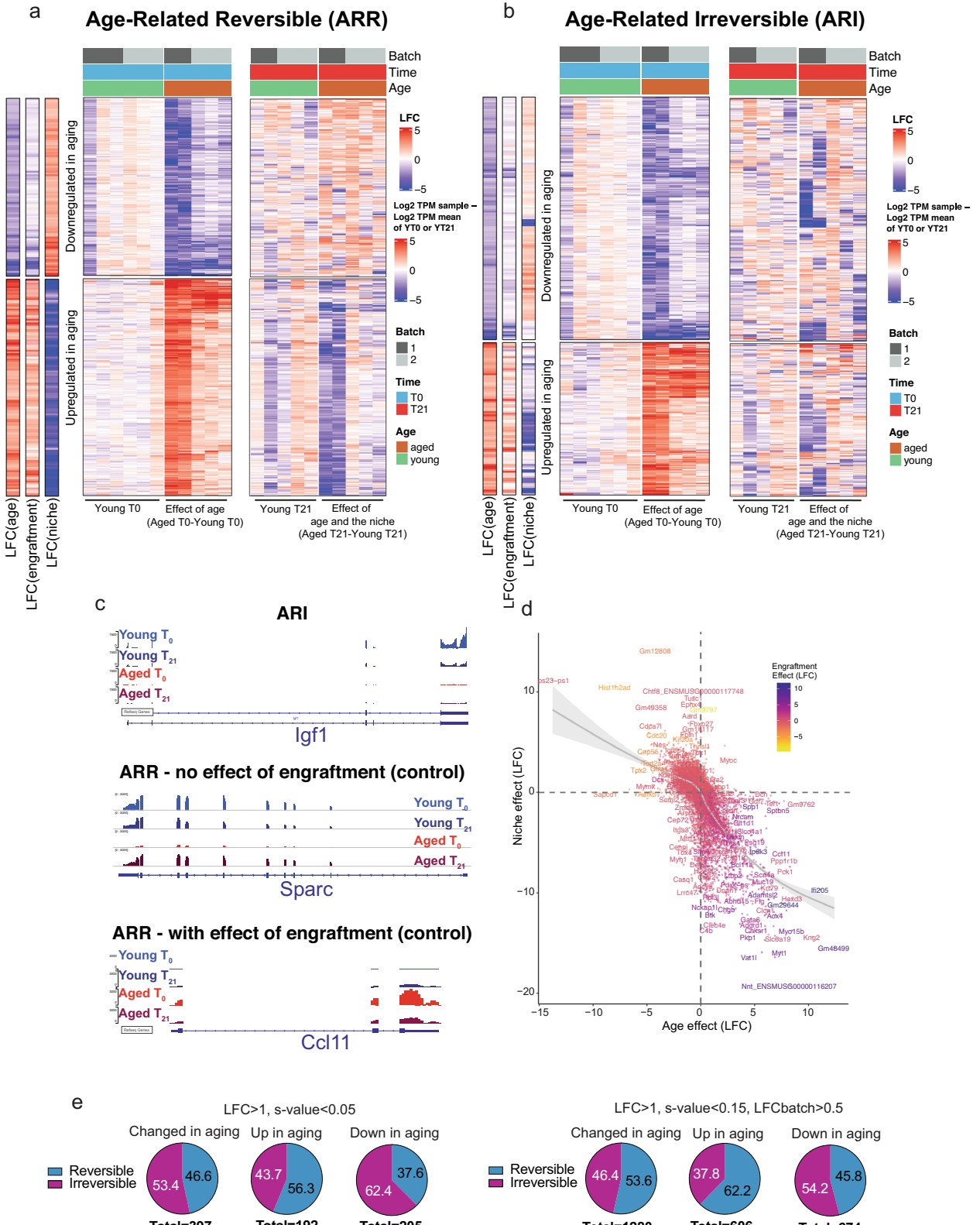

## Allogeneic MuSC transplantation into a heterologous niche

NOD-*Prkdc*^em26CdS2^*Il2rg*^em26Cd22^/NjuCrl (NCG) immunocompromised mice (Charles-River) were irradiated on their hindlimb the day before transplantation. Mice were anesthetized by IP injection of a rodent cocktail composed of ketamine (100 mg/kg)/xylazine (10 mg/kg)/acepromazine (3 mg/kg) and their hindlimb was positioned into the field of irradiation of a Multirad225 irradiation machine (Precision X-Ray)

and were given a 18 Gy dose of irradiation as described previously[59–61]. The next day, MuSCs were isolated from Tg(Pax7-EGFP)#Tajb[46] mice as described previously, and sorted into a 2% FBS/PBS mix. 10 000-20 000 freshly isolated GFP+ donor MuSCs were immediately injected into the Tibialis Anterior (TA) muscle of the previously irradiated hindlimb of host NCG mice using a Hamilton Syringe, in 2–3 separate injection sites along the TA muscle.

**Fig. 5 | Exposure to the young niche environment restores a significant portion of the age-related changes in gene expression in MuSCs. a** Heat maps of age-related reversible genes (ARRs) (moderated LFC > 1, *s*-value < 0.15, LFC batch <0.5). Left columns display the LFCs due to age (Aged T0−Young T0), engraftment (Young T21−Young T0), and the niche (Aged T21−Aged T0−LFC engraftment). The middle column displays the effect of ageing, using the mean of Young T0 samples as baseline. The right column displays the effect of the heterochronic niche, using the mean of Young T21 samples as baseline. **b** Heat maps of age-related reversible genes (ARRs) (moderated LFC > 1, *s*-value < 0.15, LFCbatch>0.5). Left columns display the LFCs due to age (Aged T0−Young T0), engraftment (Young T21−Young T0), and the niche (Aged T21−Aged T0−LFC engraftment). The middle column

displays the effect of ageing, using the mean of Young T0 samples as baseline. The right column displays the effect of the heterochronic niche, using the mean of Young T21 samples as baseline. **c** Integrative Genomics Viewer (IGV) tracks for select representative ARIs and ARRs **d** Scatterplot of the moderated LFCs of the effect of niche exposure vs the effect of age. Gray area represents the 95% confidence interval for the mean. **e** Pie charts showing the percentage of genes that are up, downregulated, or changed in in either direction in aging that are either reversible or irreversible by exposure to the young niche. Left and right groups of pie charts correspond to different threshold cutoffs for differential gene expression.

## Immunostaining of TA muscles

TA muscles were dissected tendon to tendon and rinsed in cold PBS. The isolated muscles were then fixed in 0.5% Paraformaldehyde in PBS for 2.5 h at 4 °C and then placed in 20% Sucrose (S7903, Sigma) at 4 °C overnight. Muscle samples were frozen in aluminum foil cups of Optimal Cutting Temperature (OCT) compound using liquid nitrogen-chilled isopentane and stored at −80 °C prior to sectioning by the cryostat at 8 µm interval. TA muscle cross-sections on glass slides were circled using a hydrophobic PAP pen. Cross-sections were permeabilized in 0.2% Triton X-100/0.1 M glycine in PBS for 7 min on a shaker, and then washed in PBS. Cross-sections were blocked in 3% (m/v) BSA/10 % Goat Serum solution for at least 1 h at RT in a humid chamber. TA muscle sections were then incubated with primary antibodies for PAX7 (DSHB, AB_528428, 1:100), GFP (Invitrogen, A-11120, 1:1000), and/or Laminin (Sigma, L9393, 1:750) diluted in blocking solution in a humid chamber at 4 °C overnight, then washed 3 times for 10 min with PBT (0.05% Triton X-100 in PBS). Secondary antibodies diluted in blocking solution were added for 1 h at RT in a humid chamber, followed by 3 ×10-min washes with PBT. ProLong Gold Antifade Solution with DAPI (P3695, Invitrogen) was used for mounting prior to imaging.

## Analysis of cell purity post FACS isolation

Cells were freshly sorted by FACS into a 1.5 ml Eppendorf tube containing 100 µl of 2% FBS/PBS (v:v). Freshly sorted cells were pelleted by centrifugation at 600 × *g* and resuspended in 50 µl of PBS. The cells were then plated in a well of a 12-well plate, and excess liquid was allowed to evaporate for around 10 min, or until wells appeared dry. Cells were then fixed using 3.2% PFA/PBS to the well, gently to not dislodge the cells, for 10 min at RT. After washing 3x with PBS, cells were permeabilized with 0.5% Triton X-100 for 30 min at RT. Cells were then washed 2X with 0.3% Triton X-100/PBS and blocked with 0.3% Triton X-100/0.5% BSA/PBS for 1 h at RT. Primary antibodies for PDGFRA (Cell signaling, 31745), F4/80 (Invitrogen, 124801-82), and PAX7 (DSHB, AB_528428) were diluted in blocking solution and added for an overnight incubation at 4 °C in a humid chamber. Next, the cells were washed 3X with 0.3% Triton X-100 before the addition of the secondary antibodies, diluted in blocking buffer for 1 h at RT. Finally, cells were washed 2X with 0.3% Triton X-100 before the addition of PBS with DAPI and visualized on a microscope.

## Single-cell RNA-sequencing (scRNA-seq)

MuSCs, macrophages and FAPs were isolated from three young (5 weeks old) and four aged (23.5 months old) mice by FACS, as described previously, and immediately processed for single-cell RNA-sequencing (scRNA-seq) using Chromium Single Cell 3' Reagent Kits (v3 Chemistry) using 10x genomics technology. In order to maintain ratios, all cells were sorted into a single tube and to aim for a total of 2500 captured cells for sequencing per biological replicate. The viability of cells was 85.5% on average, with a capture range of 2437 to 2835 cells. Quality control and processing steps can be found at https://csglab.github.io/transcriptional_reprogramming_muscle_cells/.

## SMART-seq library preparation

MuSCs were sorted by FACS directly into 9 µl of nuclease-free water containing 1 µl SMART-Seq Reaction Buffer, which is composed of 10x lysis Buffer supplemented with RNase inhibitors. The volume was brought to 11.5 µl and libraries were processed using the SMART-Seq HT library preparation kit (634456, Takara Bio) as described[47,62]. cDNA libraries were then purified using Ampure XP beads (A63880, Beckman Coulter) at a 1:1 (v:v) ratio and quantified using the Quant-iT PicoGreen dsDNA Assay kit (P11496, Invitrogen). 0.15 ng of cDNA in a 1.25 µl volume was used as input for Nextera XT (FC-131-1024, FC-131-2001, Illumina) tagmentation, as described[63,64]. Sequencing libraries underwent Ampure XP size selection using a 1:0.85 (v:v) ratio, and an aliquot was used to verify size on an agarose gel stained with GelGreen dye (41005, Biotium). Final libraries underwent quality control verifications using a bioanalyzer and were sequenced on an Illumina NextSeq500.

## Assay for transposase-accessible chromatin using sequencing (ATAC-seq)

Low input ATAC-seq (Assay for Transposase-Accessible Chromatin using sequencing) was performed according to the OMNI ATAC-seq[65] protocol. Briefly, 5000 MuSCs were FACS sorted into the lysis buffer (10 mM Tris-HCl pH 7.5, 10 mM NaCl, 3 mM MgCl2, 0.1% Tween-20, 0.1% NP-40, 0.01% Digitonin) and incubated for 5 min on ice and subsequently 3 min at room temperature (RT). Cells were then washed with 100 µl of wash buffer (10 mM Tris-HCl pH 7.5, 10 mM NaCl, 3 mM MgCl2, 0.1% Tween-20) and spun at 800 g for 10 min. The pellet was resuspended in the transposition mixture at a total volume of 10 µl (5 µl of Tagment DNA (TD) buffer, 3.2 µl PBS, 0.89 µl Tn5, 0.1% Tween-20, 0.01% Digitonin and 0.75 µl water). Transposition was performed for 20 min at 37 °C while shaking the tubes every 5-7 min. The DNA was then purified using column purification as described (Qiagen, QIAquick PCR Purification Kit Cat: 28104). The purified DNA was then PCR-amplified using Q5 DNA polymerase for 12 cycles with the Illumina Nextera XT adapters. The amplified DNA was then size selected and purified with Ampure XP beads at a 1:0.85 (v:v) ratio. The quality control of the final libraries was performed by bioanalyzer and paired-end sequencing was performed on NovaSeq 6000 Sprime paired end (PE 150 bp).

## Genomic DNA extraction

Genomic DNA was extracted using phenol/chloroform/isoamyl alcohol solution (Invitrogen #15593-031). Briefly, primary myoblasts derived from in vitro expansion of freshly sorted MuSCs were pelleted at 600 × *g* for 10 min and snap frozen in liquid nitrogen before use. When all samples were collected, cell pellets were thawed on ice and lysed in 400 µl lysis buffer (40 mM Tris-HCl pH 8.0; 1% (vol/vol) Triton X-100; 0.1% (vol/vol) SDS; 4 mM EDTA (pH 8.0) and 300 mM NaCl). Cell lysates were incubated at 37 °C for 45 min with 10 µl RNase A (Invitrogen, cat#12091-021), followed by a 1 h 15 min incubation at 45 °C with 10 µl of Proteinase K (10 mg/ml Millipore, cat#70663). One volume of phenol/chloroform was then added to each lysate, and samples were vortexed vigorously and centrifuged at 2110 × *g* for 3 min. The aqueous

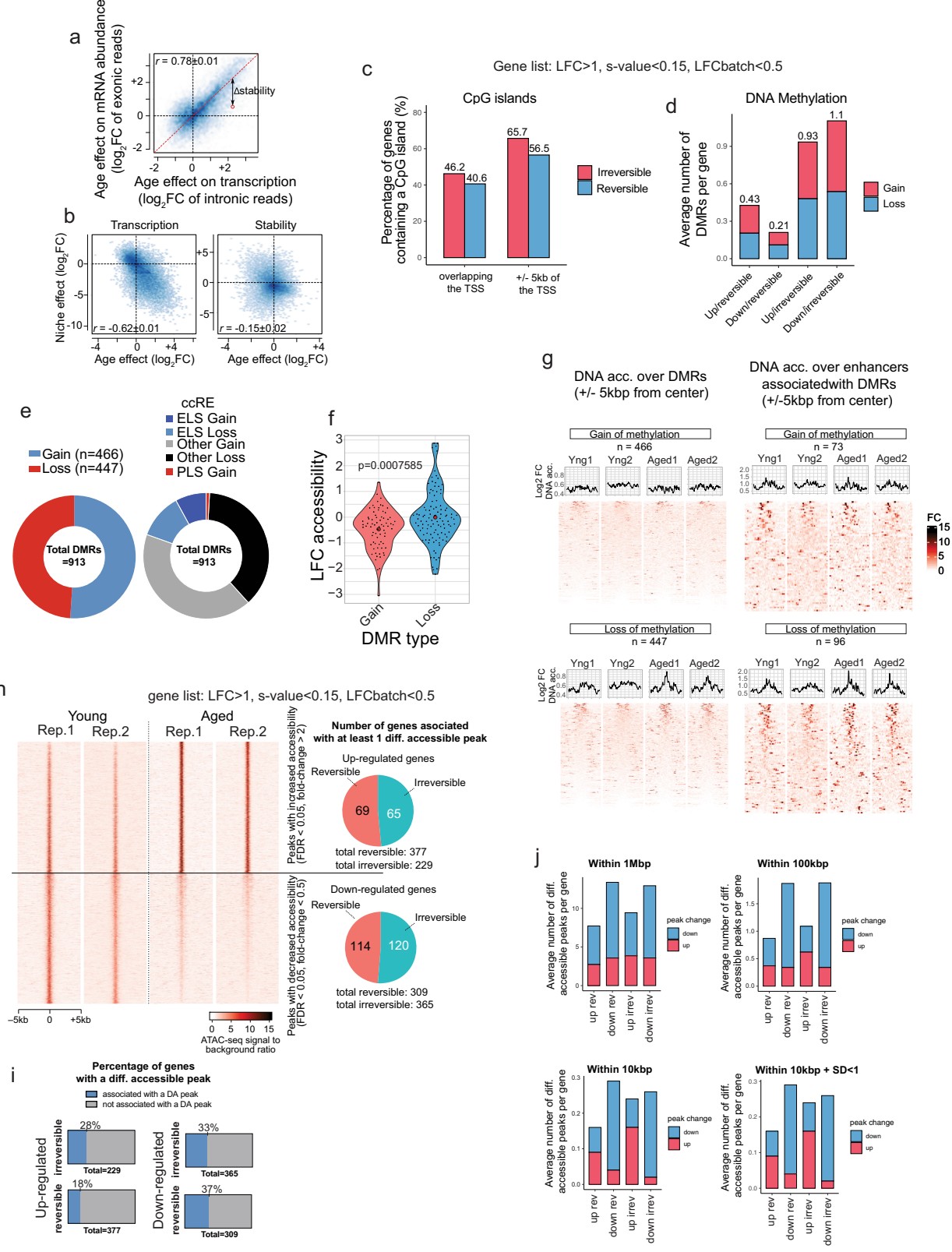

layer was retained and mixed with 2 volumes of 100% ethanol (EtOH). Samples were snap frozen in liquid nitrogen for 3 min, thawed on ice, and centrifuged for 20 min at 2110 × *g*. The DNA pellet was washed 3X with 750 µl of 75% EtOH, spinning each time 2110 × *g* for 12 min. The DNA pellet was then air-dried and resuspended in TE buffer.

**Whole-genome bisulfite sequencing (WGBS) library preparation**
Genomic DNA was sheared using COVARIS, and libraries were constructed using the NxSeq AmpFREE low DNA Library Kit by Lucigen. Bisulfite conversion was done using the EZ-96 DNA Methylation-GoldTM MagPrep kit (D5042, Zymo Research).

**Fig. 6 | The epigenetic state dictates the responsiveness to heterochronic niche exposure of genes with altered expression in aging. a** Analysis of RNA stability. Scatterplot showing the age effect on transcription, as measured by changes in intronic read abundance between aged and young cells compared to the age effect of mature RNA abundance, based on exonic reads. Values along the diagonal represent genes that do not show significant difference in RNA stability between MuSCs from young and aged mice. **b** Scatterplots showing the effect of age compared to the effect of the heterochronic niche on transcription (left) and RNA stability (right). **c** Quantification of the percentage of reversible and irreversible genes containing a CpG island overlapping with or within 5 kb of their TSS. **d** Bar plot showing the average number of DMRs per gene for each gene category. **e** Pie-chart showing the proportion of age-related DMRs corresponding to a gain or loss of methylation (left). Pie Chart showing the proportion of identified age-related DMRs associated with cCREs (±300 bp) that are located within Enhancer-like sequences (ELS), promoter-like sequences (PLS), or other regions of the genome (right). **f** Violin plots showing the LFC of MuSC chromatin accessibility and age-related DMRs (two-tailed $t$ test). **g** ATAC-seq pileup analyses showing chromatin accessibility either over DMRs (±2KB) or over enhancers associated with DMRs in young and aged MuSCs. **h** Left: Heatmap of differentially accessible ATAC-seq peaks, including peaks with increased accessibility in aged cells (up) and those with decreased accessibility (bottom). Right: Pie-charts showing the overlap of differentially accessible peaks with different gene categories. The top pie chart shows the number of reversible/irreversible age-upregulated genes that are associated with at least one age-upregulated peak. The bottom pie chart shows the number of reversible/irreversible age-downregulated genes that are associated with at least one age-downregulated peak. **i** Bar charts showing the percentage of genes in each category that overlap with differentially accessible peaks. **j** Bar plots showing the average number of differentially accessible peaks per gene in each category, within 1 Mbp, 100 kb, and 10 kb from the gene TSS (LFC > 1, FDR threshold = 0.05). The bottom right panel involves only genes with the highest reproducibility (standard deviation <1) between samples and batches.

## Analysis of differentially methylated regions (DMRs) after whole-genome bisulfate sequencing

Quality check and adaptor trimming of raw reads was performed by trim-galore (Default setting, Phred score = 20)[66]. Reads were aligned to the mouse reference genome (mm10) using bismark (internal bowtie2 aligner's default settings and allowing one mismatch)[67]. Extraction of methylation calls was done by ignoring the first 2 base pairs from the 5' end of read 2 to avoid known experimental-introduced bias[68].

The DMRs between young and aged samples, were identified with DMRcaller's (R/Bioconductor) function computeDMRsReplicates[69]. DMRcaller implements a Beta regression to test the difference in methylation levels between conditions in regions of 100 bps with at least 4 CpG sites. Only regions with a minimum methylation proportion difference of 40% are reported. *P*-values are adjusted for multiple testing using Benjamini and Hochberg's method (adjusted *p*-value < 0.01)[70].

## Bulk RNA-seq data analysis

Transcript-level abundances were imported into R and collapsed into gene counts using the *tximport* package[71]. To estimate the effects of the experimental conditions, we used *DESeq2*[72] to fit the following gene-wise model: *count- time + age + time:age*, where the variable *time* denotes the effect of engraftment, the variable *age* denotes the effect of aging, and the interaction variable *time:age* represents the reprogramming effect on genes in the aged niche (i.e. the residual effect beyond that of engraftment or aging that explains observed expression of a given gene). For the purposes of ranking and visualization, we used the *apeglm* shrinkage approach[73].

## Single-cell RNA-seq data analysis

We used *Salmon* and a*levin*[74,75] to quantify spliced mRNA and unspliced pre-mRNA abundances from scRNA-seq data. To create a spliced-intronic Salmon quantification index, we used *eisaR*[76] to generate a combined FASTA file containing exonic and intronic sequences from the mouse Genocode M24 transcriptome and genome reference files. The introns were defined using the "collapse" option in which transcripts of the same gene are first collapsed by taking the union of all the exonic regions within a gene and labeling the remaining non-exonic parts as introns. In contrast to the alternative "separate" strategy in which intronic regions are extracted for each transcript separately, the collapse strategy treats an ambiguous read to be more likely to have come from an exon of a given transcript than from an intron of alternate transcript that happens to overlap that exonic region[76]. A 90 bp flanking sequence was added to each side of each extracted intronic region (equal to the length of Read 2 in Chromium Single Cell 3' Gene Expression library with v3 chemistry) in order to quantify intron-exon reads as unspliced pre-mRNA abundances. To exclude reads coming from intergenic regions of the genome, we adopted the

selective alignment (SA) strategy by providing Salmon the complete genome sequence as a decoy sequence[77].

The alevin quantifications were imported into *R 4.0.0* using the *tximeta* package. The abundances were split into a spliced and unspliced abundances matrices to be used for downstream analysis such as RNA velocity. We identified and removed low-quality or potentially multiplet cells based on whether they had high percentage of mitochondrial reads (>15%), low library sizes, low number of genes detected, or whether they tend to be observed as outliers with respect to the distributions of spliced reads, unspliced reads, or the fraction of unspliced reads in each cell (see the analysis notebooks for details: https://csglab.github.io/aging_muscle_niche/pages/notebooks.html).

To cluster cells, we used UMAP[78] followed by the Walktrap community finding algorithm[79] to reduce the dimensionality of the data and pull together similar cell populations into separated clusters which can then be labeled. To annotate the main clusters, we used the expression of *Adgre1*, *Ptprc*, and *Itgam* as markers of macrophages, *Pax7* as a marker of muscle stem cells (MuSCs), and *Pdgfra* as a marker of FAPs. The initial clustering showed the presence of three large main clusters corresponding to the three cell types along with four small populations whose expression profiles suggest them as being Schwan and smooth muscle cell clusters respectively (Supplementary Fig. 1e-g). The four small cell clusters were ignored in downstream analysis. Next, the seven samples were batch normalized and corrected for batch effects using fast mutual nearest neighbors (MNN) correction[80]. The clustering approach was repeated independently on the MNN corrected space of the MuSCs, FAP, and macrophages respectively in order to determine the internal heterogeneity of each cell type cluster. As a result, MuSCs were partitioned into three clusters, the FAPs into two clusters, and the macrophage cells into one cluster only.

We ran differential gene expression analysis separately for each cell type. We discarded genes that had low counts, that had low mean expression across the subclusters making up the cells of each cell type, and that were characterized by severe batch effects. We used ZINB-WaVE[81] to compute gene and cell-specific observational weights that can be used to unlock *DESeq2*[72] for the differential analysis of single-cell RNA-seq data. For data of each of the three cell types, we independently fit the following gene-wise model: *count~ −1 + cluster_condition + sample*, where the variable *cluster_condition* denotes the mean expression of cells in each cluster and age condition, and the variable *sample* denotes the batch effect of each sample. We manually created the design matrix such that the sample effects of one of the aged samples and one of the young samples are removed in order to fit a statistical identifiable model. We tested several contrasts to assess several hypotheses of differential mean gene expression across the different subclusters. For the purposes of ranking and visualization, we used the *ashr* shrinkage approach[82] to preserve the size of estimated large LFC and compute *s*-values (the estimated rate of false sign).

### Integration of bulk and single-cell RNA-seq data

Using the moderated LFCs generated by models fit on both RNA-seq data modalities, we are able to examine the pattern of concordance of differential gene expression across the multiple datasets. More specifically, the moderated LFC model effect estimates for the bulk data and the muscle stem cells (MuSCs) single-cell data were joined together such that we retain only genes that were not discarded from the scRNA-seq data. The LFCs were used for plotting Supplementary Fig. 9g.

### Geneset Enrichment Analysis (GSEA)

We used the R package hypeR[83] to conduct hypergeometric tests to determine which respective set of statistically significant differentially regulated genes are over-represented in the 50 pre-defined hallmark gene sets listed in the Molecular Signatures Database (MSigDB)[84]. The threshold for significance was set at 10-8 for the *s*-values across the contrast effects accompanied with an absolute value cutoff for the moderated LFCs between 0.5 and 2 depending on the distribution of effect values for each contrast. The six sets of significant genes for MuSCs are the following: (1) genes that are downregulated in aging both in bulk and single cell, (2) genes that are upregulated in aging both in bulk and single cell (scRNA-seq), (3) genes that are upregulated in MuSC2 with respect to MuSC1 cluster, (4) genes that are upregulated in MuSC1 with respect to MuSC2 cluster, (5) genes that are upregulated in MuSC3 with respect to young cells in both MuSC1 and MuSC2 clusters cluster, (6) genes that are downregulated in MuSC3 with respect to young cells in both MuSC1 and MuSC2 cluster. For FAPs, there were four sets: (1) genes that are downregulated in aging (2) genes that are upregulated in aging (3) genes that are upregulated in FAP 2 with respect to FAP1, (4) genes that are upregulated in FAP1 with respect to FAP2.

### ATAC-seq pre-processing analysis

The ATAC-Seq data was processed using the ENCODE ATAC-seq pipeline (https://github.com/ENCODE-DCC/atac-seq-pipeline). The pipeline consists of adapter read trimming, read alignment, post-alignment read filtering and de-duplicating, calling peaks, creating signal tracks, generating quality control reports, and running Irreproducible Discovery Rate (IDR) analysis[85]. The full specification of the pipeline is detailed on the website (https://github.com/ENCODE-DCC/atac-seq-pipeline). For downstream analysis the generated IDR optimal peaks were used.

### Analysis of differentially bound ATAC-seq peaks

Peaks from young and aged conditions were merged from the IDR pipeline. Peaks with summits within 200 bp were combined, a new summit was set in the middle. The peaks' start and end were redefined with a range of 500 bp from the summit.

We created a count matrix by extracting ATAC-seq read counts from peak and background (+/− 10kbp around the peak) regions for all four samples, peaks and samples correspond to rows and columns respectively[86]. We used DiffRAC and the count matrix to identify significant differentially accessible peaks (adjusted *p*-value < 0.05 and LFC >1)[50]. Differentially accessible peaks were associated to genes using GREAT[87] with the default parameters for the Basal plus extension mode, where a proximal gene regulatory domain is defined as 5kbp upstream and 1 kb downstream of the gene TSS, and a distal domain of 1 Mb.

### Analysis of differential mRNA stability

Differential mRNA stability was inferred as previously described[50]. Briefly, RNA-seq reads were mapped to exonic and intronic regions using annotations acquired from Ensembl GRCh38 version 87. DiffRAC was then used to decouple transcriptional and post-transcriptional effects and infer changes in mRNA stability.

### Differential analysis of cell-cell interactions

To identify the cell-cell interactions occurring in young and aged mice, we used the scDiffCom package in R[44]. To run the software, the Single Cell Experiment object containing the scRNA-seq data was converted to a Seurat object and provided as an argument to the scDiffCom method using the mouse genome setting. We defined the conditions as aged and young mice samples and grouped Emitter (ligand source) and Receiver (receptor source) cells by cell type (FAP, MuSC and Macrophage). The calculated interaction scores were classified under 4 categories in aged samples relative to young samples: UP-regulated for log-fold changes above log2 (1.5) and adjusted *p*-values below 0.05, DOWN-regulated for log-fold changes below −log2 (1.5) and adjusted *p*-values below 0.05, FLAT for absolute log-fold changes below log2 (1.5), and NSC (not significant) for absolute log-fold changes above log2 (1.5) with adjusted *p*-values above 0.05. We then calculated the over-representation scores of the Emitters, Receivers and Emitter-Receiver pairs and plotted them as a network.

### Transcription factor footprinting

ATAC-seq read biological replicates were merged per condition (young and aged). Peaks per condition were called with MACS2 (−*nomodel,−shift −100, and−extsize 200*) and merged[88,89]. Tn5 bias, prediction of TF binding sites, and visualization of differentially bound TFs was done with TOBIAS[90]. Repetitive and ENCODE's blacklisted regions were excluded from the analysis[91] (Smit, A., Hubley, R. & Green, P. RepeatMasker. http://www.repeatmasker.org/ (2013). JASPAR motifs (non-redundant, vertebrate) were used as reference[92].

### Animal care

Mice were housed at a temperature of 21° with 20% humidity, with a light/dark cycle of 14 h light/10 h dark. All animal protocols and procedures carried out were approved by the McGill University Animal Care Committee (UACC).

### Reporting summary

Further information on research design is available in the Nature Portfolio Reporting Summary linked to this article.

## Data availability

The Next-Generation Sequencing (NGS) data generated in this study have been deposited in the NCBI Gene Expression Omnibus database under super series GSE171998. Individual accession numbers for WGBS, scRNA-seq, RNA-seq, and ATAC-seq are GSE171604, GSE171794, GSE171997, and GSE171534, respectively. The processed sequencing data generated in this study are provided in Supplementary Data 1–5.

## Code availability

Reproducible analysis notebooks are available on the GitHub project's website at https://csglab.github.io/transcriptional_reprogramming_muscle_cells.

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

## Acknowledgements

The authors thank Christian Young at the Lady Davis Institute for Medical Research—Jewish General Hospital—core facility for help with fluorescence-activated cell sorting (FACS). We thank Dr. Shahragim Tajbakhsh from Institut Pasteur, Paris, France for kindly providing us with Pax7-nGFP mice. We thank Dr. Roderick R. McInnes at the Lady Davis Institute for Medical Research—Jewish General Hospital and the McGill University Department of Human Genetics for careful review and editing of our manuscript. This work was supported by a Disease team grant from the Stem Cell Network, Project grant PJT-156087 from the Canadian Institute of Health Research (CIHR), and a Richard & Edith Strauss Canada Foundation grant to VDS, CFI-MSI fund 35444 and CFI funds 33408 and 40104 to JR, as well as NSERC Discovery grant RGPIN-2018-05962, CIHR grant PJT-155966, and Compute Canada resource allocations to HSN.

GP is supported by training scholarships from the Canadian Institutes of Health Research (CIHR), the Fonds de recherche du Québec – Santé (FRQS), and Oncopole. TL is supported by a Vanier Canada Graduate Scholarship and an FRQS doctoral fellowship. HSN holds a Canada Research Chair funded by the CIHR.

## Author contributions

Conceptualization: V.D.S.; Methodology: F.L., V.D.S., R.F., K.S., A.H.C., H.S.N.; Investigation: V.D.S., F.L., R.F., K.S., D.B., A.H.C., T.L., A.O., H.S.N., G.P., T.P.; Visualization: F.L., V.D.S., R.F., H.S.N., A.H.C.; Funding acquisition: V.D.S., H.S.N.; Project administration: V.D.S.; Supervision: V.D.S., H.S.N.; Resources: V.D.S., A.J.A., J.R., C.C., H.S.N.; Software: R.F., A.H.C., T.P., G.P., H.S.N.; Writing—original draft: FL, VDS; Writing—review and editing: FL, VDS, RF, DB, KS, AJA, HSN.

## Competing interests

The authors declare no competing interests.
