## [Peer Review File · Nature Communications]

Transcriptional reprogramming of skeletal muscle stem cells by the niche environmentReviewer #1 (Remarks to the Author):

The article by Lazure et al is an interesting manuscript that describes profiling of muscle stem cells (MuSCs) in several different conditions (at different ages and before and after transplantation into immunocompromised mice). The authors first assess changes in the transcriptomes of single MuSCs using single-cell RNA sequencing after FACS isolation. Next the authors isolate MuSCs from young and aged mice and perform transplantation into immunocompromised hosts. To evaluate changes in the transcriptome as a result of transplantation, muscle stem cells were isolated and profiled again by gene expression profiling. Last, the authors performed chromatin accessibility measurements and DNA methylation measurements using whole-genome bisulfite sequencing. Analysis of the datasets was performed and the authors conclude there are variations in the epigenome with aging. Overall, the results are interesting but many of the generated datasets already exist dampening enthusiasm for value added to the field. Additionally, there is little to no validation of the datasets, several experiments are statistically under-powered and many interpretations would need to be significantly scaled back. The primary limitation of the manuscript is that there are no validation experiments performed, and as written, the manuscript seems disjointed and a collection of datasets rather than a linear story. Major concerns are listed below for each of the figures and sections.

Figure 1a-1d: The authors use FACS to isolate MuSCs, fibro-adipogenic progenitors and macrophages and claim changes in the number of isolated cells. $n=3$ and $n=4$ replicates may be acceptable to perform statistical analysis of this population numbers but this was not performed. Additionally, FACS can be highly variable for enumerating cellular fractions and the authors should validate their claim of decreases in cell numbers through in situ staining. Moreover, given the claims in the manuscript about potential cellular crosstalk, the authors could quantify cellular distances between these cell types to determine if they are physically close and distances are altered with age.

Figure 1e-1j: The authors appear to have a single replicate for scRNA-Seq pooled from 3 young mice and 3 aged mice. This may be acceptable but there is no quality control presented for the single cell datasets (genes detected, UMIs detected, etc). As shown, it is unclear whether the resulting integrated nearest neighbor graph is accurate and there are three separate cell states or whether the authors' downstream analyses have produced this type of result. What is the nearest neighbor classification accuracy for these cell states if the authors used different integration strategies and hyperparameter choices? Given the availability of single cell datasets from mouse muscle stem cells at different ages, the authors could consider integrating their datasets with others to determine if population differences the authors are observing are also present. Inspection of other datasets such as from Tabula Muris Senis for example does not show segregation into 3 clusters. Are the differences observed and described here explained by differences as a result of experimental preparation?

Figure 2: The authors could experimentally demonstrate what physiological differences there are between the 3 groups of MuSCs (proliferative differences, engraftment efficiencies, surface markers or transcription factors expressed, etc). As presented, it is speculative if the observed differences are meaningful or an artifact of the bioinformatics processing. Additionally, many of the genes discussed have already been profiled in MuSCs and FAPs diminishing novelty of this set of results. The statement that aging impinges on common gene networks and pathways is not experimentally demonstrated with the data presented. The authors could validate their findings with RNAScope or immunohistochemistry additionally to determine if this statement is true. Lastly, its very difficult to see some of the differentially expressed genes in Figure 2.

Figure 3: The authors used a transplantation model to decouple the effects of signaling from the niche into MuSCs and performed gene expression profiling 21 days after transplantation. This strategy is interesting but given the tremendous amount of cellular death from transplantation, it seems as though the strategy selects for a subset of cells.

Additionally, recent evidence (Liu et al, Cell Stem Cell 2018) has shown that aged activated MuSCs are more prone to cellular death than young activated MuSCs. Thus, the transplantation assay seems to “select” cells rather than assaying the true diversity that exists in aged animals. As an alternative, the authors could perhaps do validation of what state the MuSCs that remain are in (are they quiescent, still activated?). This would be helpful to argue if differences observed are because MuSCs are mostly still activated or have returned to quiescence at the time profiled. The authors could also consider increasing their sample size as n=2 seems low given the variance plotted in their PCA plots and Spearman correlations did not seem to be reported. The authors could also somehow try to connect their datasets to the single cell datasets (such as staining to determine if transplantation results in enhancement of interactions between cell types or decreases). Lastly, a recent manuscript was published that (Evano et al PLoS Genet 2020) performed similar studies whereby MuSCs were profiled using RNA-Seq and DNA methylation before and after transplantation but the authors did not cite this paper.

Figure 4: Similar to other experiments above, many of the identified variations from the experiment are not validated and the conclusion of the experiment, which is that “the niche is a principal regulator of the MuSC transcriptome,” and that “a significant portion of the transcriptome of MuSCs can be reprogrammed back to a youthful state by exposure to a young niche milieu” seems premature. If genes are reversibly changed, can the authors validate these results first at the protein level and then by rescue in aged animals? The authors don’t need validate many but 1-2 predicted targets seems reasonable. The authors could also use their single cell datasets to identify potential signaling cells that are altered and which may be deleterious and which may advantageous for MuSCs.

Figure 5: The authors claim about differential RNA stability should be validated with an orthogonal assay for several transcripts through qPCR. If there are differences manifesting from deposition of cytosine methylation between young and aged MuSCs but the authors are performing the assay on myoblasts which have different methylation profiles, the authors should discuss this aspect in their results section. As written, this section seems underdeveloped and should be connected to enzymes that deposit remove or deposit these modifications. The enrichments of the ATAC-Seq assay look to be strong, but the authors may consider using the Irreproducibility Discovery Rate (IDR) framework to look at agreement between peak calls. The comment that genes that are niche-responsive are transcriptionally fluid seems like an over-statement and the data do not effectively reflect this claim. Given there are several ATAC-Seq (Garcia-Prat et al, Nature Cell Biol. 2020 & Shcherbina et al, Cell Rep 2020), and DNA methylation datasets on young and aged MuSCs (Hernando-Herraez et al, Nature Comm 2019, Evano et al PLoS Genet 2020), the authors could consider integrating these datasets and observe if the variations in their datasets are unique or similar.

Reviewer #2 (Remarks to the Author):

The manuscript by Lazure et al. address a significant question in stem cell biology i.e. whether the age-related decrease in stem cell function is due to ‘intrinsic’ changes or the ‘extrinsic’ cues from an aged niche. They utilize single cell RNA-seq data and analyses to dissect out the relative contributions of these ‘intrinsic’ and ‘extrinsic’ changes and come to the really interesting conclusions that the ‘intrinsic’ changes appear to be largely driven by epigenetic changes in the muscle stem cells.

The study is interesting and significant, however there are some concerns/suggestions as follows that need to be addressed and revised:

1. The authors have nicely delineated the different clusters of MuSC that alter with aging in abundance; it had not been described how the old MuSC clusters change in abundance

or quality upon transplantation in the young niche. For example, is the young niche capable of reprogramming old MuSCs into the cluster 3 MuSCs that are lost with age? How do the abundance of MuSC 1 and 2 change in this experiment?

2. It would be very informative to analyze separately how the MuSC 1 and 2 clusters respond to the young niche? Are there pathways that are specific to one vs another?

3. The rationale for using myoblasts derived from MuSCs for the Methylation analyses is not clarified-it would be much more relevant to directly compare the epigenetic patterns and gene expression data in the same cell types i.e. MuSCs in which the RNA-seq and ATAC seq are done. This is an important point since it affects the central conclusion of the study.

Reviewer #3 (Remarks to the Author):

In this manuscript, the authors compared gene expression profiles between young and old muscle stem cells (MuSC) via single-cell RNA-seq (scRNAseq) and demonstrated that there are three clusters (clusters 1, 2, and 3). Cluster 1 (MuSC1) is more prone to age-related loss, whereas the cells in Cluster 2 (MuSC2) display enhanced retention with age. Cluster 3 (MuSC3), which is characterized by genes involved in MuSC activation such as MyoD1 and Cdkn1c is completely lost in aged mice. MuSC transplantation of young and aged cells into heterochronic host muscles demonstrated that the expression of half of the age-altered genes could be restored by exposure to a young niche environment. Unaltered genes exhibit altered chromatin accessibility shown by ATACseq and Methylome. By contrast, the majority of age-related altered genes are not epigenetically encoded and restorable by exposure to a young niche environment. Therefore, the authors concluded that the stem cell niche might restore the age-related gene expression dependent on chromatin accessibility.

This is an excellent and unique approach for age-related gene expression and alteration by the age-associated niche environment. Before publication, the authors should address the following issues.

1. While MUSC3 is mentioned as an activated MuSC population, the characteristics of MUSC1 and MUSC2 are not so clear. Would you please explain which kind of clusters they are and which representative genes with both clusters should be indicated?

2. In Supplementary Table 2, sample numbers of scRNAseq for MuSC transplantation of young and aged cells into heterochronic host muscles include Aged T0 (x2), Aged T21 (x3), Young T0 (x3), and Young T21 (x2). This reviewer feels triplicated sample number is required for each group since some gene expressions, especially genes for key myogenic factors and aging-associated genes, display significant variability, maybe due to the sample preparations or timing. For example, Myog: (Aged T0: 2.0275, 4.5914), (Aged T21: 0.036562, 39.6274), (Young T0: 1.0838, 20.6157, 194.1455), (Young T21: 25.5482, 72.7847).

3. Extended Data Figure 4e shows 3x Young T21 samples, but in Supplementary Table 2 and Extended Data Figure 4f shows 2x Young T21 samples. Would you please explain this discrepancy?

Reviewer #4 (Remarks to the Author):

The manuscript from Lazure et al investigates the transcriptional and epigenetic landscape of muscle stem cells. The work focuses on the process of stem cell aging, and within this context on the role of the stem cell niche that is well known to play a crucial role in stem cell maintenance and cell quiescence. By using an elegant experimental design and sorting strategy, as well as screens on the transcriptome, chromatin accessibility, and DNA methylation level, Lazure and colleagues were able to unravel differentially regulated entities respectively and to classify the given changes into age dependent, and stem cell niche dependent changes. Finally, a transplantation approach

with young and aged stem cells delivered into radiated young mice allowed for the investigation of reversible gene expression patterns in dependency of the young niche. Of note, Lazure and colleagues found distinct subtypes of stem cells and FAPs by SC-RNAseq and a unique subpopulation of MuSC exclusively given in young mice. Further, they were able to define a reversible and irreversible gene panel in dependency of a young niche. Finally, they found evidence for epigenetically coded (DNA methylation, accessibility) age-deregulated alterations to be less responsive to the influence of the niche.

Comments to the authors

Major

1. The authors use cell sorting followed by 10x based SC-RNAseq to investigate cell subtypes and subtype proportions for MuSC, FABs and Macrophages. In case of MuSC, three and two cell clusters for young and aged MuSC are defined respectively (fig1e, fig2b/g). This is for sure a very interesting finding, and the reader is eager to learn about the potential role of these subtypes. The missing cluster three in old MuSC is of course of particular interest. Unfortunately, a more comprehensive characterization of these clusters according to widely accepted concepts of MuSC regulatory circuits as e.g. described in Tierney/Sacco 2016 (<https://doi.org/10.1016/j.tcb.2016.02.004>) is critically missing. In particular, a closer look on MuSC cluster 3, missing in aged mice, and its potential role in MuSC self-renewal would definitely strengthen this manuscript. For example, Myf5 is often used to define a long term self renewing MuSC population, and its expression is shown in ext fig1 i. Can one state that MuSC cluster 3 expresses low levels of Myf5 compared to Cluster 1 and 2 ?

Along this line, fig2 presents Cdkn1c as marker for cluster 3, known to be activated in induced MuSC starting to become myoblasts. As it stands now, the presented data is very interesting, but it opens up the room for a multitude of obvious questions that are not answered or discussed.

2. The SMART RNAseq from engrafted aged MuSC is compared and characterized in fig4 /ext4/ext5, and the author indicate a panel of reversible genes that probably push back the aged MuSC to a more young phenotype. Of course it would be interesting to know if these genes (or some of them) overlap with the "missing" cluster 3 of MuSC as shown in fig2b. It is clear that a SC approach is not possible due to low cell numbers from engrafted animals, however, a bioinformatics approach for bulk RNA deconvolution such as given with cybersortx (Newman Lab Stanford) might help to answer this question.

3. Figure 5 deals with genome wide DNA methylation data and chromatin accessibility based on ATACseq. On line 206-208, the authors state that they found a significant over-representation of irreversible upregulated genes among loss-of-methylation DMRs, and I assume that Figure 5e and ext. table 2 show the same data. It is illustrated that all combinations of up/down reversible/irreversible contain approximately the same average number of loss and gain DMRs. However, the complete analysis is based on a small number of overlapping genes, namely 15 and 17 respectively (ext. table 2). In this context it is not clear what is shown exactly. For example looking at the down/reversible bar (second from left), one can observe gain and loss at about 50% from the total height of the bar, that reflects ~ 0.9 DMRs per gene, but in table ext 2 this column indicates 0 and 1 gene.

4. In figure 5i, the authors show two sets of ATAC-seq peaks - increased and decreased accessibility by age, in total ~ 7560 peaks. On the right side, the overlap with the up- and down-regulated genes is shown, as well as an enrichment of the irreversible set of genes in both sets of peaks. However, the analysis assumes that gain-of-accessibility only leads to upregulation of genes. For completeness, it would be great if the authors would also show the overlap of loss-of-accessibility with up-regulated genes and gain-of-accessibility with down-regulated genes. I assume that the numbers in principle should be given in ext. table 3, but I was not able to find the gene numbers given in the pie chart.

5. Having a closer look on ext table 3, one also get the impression that for upregulated peaks not only irreversible genes are significant, rather than all upregulated genes. In case of the downregulated peaks and downregulated genes the reversible genes seems to be more significant than the irreversible, that do not fit to fig5i at all. Please

clarify.

6. Along this line, ext. fig. 6e shows the majority of ATAC peaks to be located in proximity to genes (promoter <1kb, introns, exons, UTR). In addition, ext. fig. 6a indicates ~50% of all TSS regions (and probably promoter/gene bodies, scale not indicated) to be accessible. Having these numbers, I'm wondering why the number of genes up/downregulated overlapping a differential peak is "just" in the range of ≤ 100 (fig 5i right) each. Does this mean the majority of differentially regulated genes is in closed chromatin, or in not changing chromatin regarding accessibility? If this is the case, can one hold generalized terms such as "that changes in chromatin accessibility and DNA methylation impede transcriptional restoration by niche exposure"?

7. Fig 5j gives a scheme on chromatin accessibility and genome responsiveness to the niche, in dependency on up/down regulation of genes. As it is given now, the scheme indicates that down regulated genes per se have a higher degree of chromatin accessibility. Further once can conclude that all upregulated genes are more responsive to a niche than downregulated genes. Is this intended by the authors? I recommend to split the figure or find a different way of visualization.

8. While it might not be possible to perform additional epigenetic screens in the engraftment-model (also due to the low amount of recovered MuSC), it would be beneficial if the authors would at least discuss the limitations of comparing natural epigenetic markers of aging with the change in niche environment used for the RNA-seq experiments.

9. Figure 5h illustrates ATAC signals at DMR positions surrounded by a 10kb window, with one panel indicating gain of methylation spots, and a second panel indicating loss of methylation spots (left panel). It is not clear to me what can be concluded from this visualization. One can speculate that the overall chromatin accessibility change in young cells is smaller than in aged cells? Why did the author choose a 10KB window, I would expect this size to include further CpGs. Did the authors follow an approach that combines individually changed CpGs to annotated regions such as promoters or tiles in order to find larger consistently changed DNA regions regarding methylation?

10. Regarding the ATACseq data analysis as it stands, I feel the authors miss to use this very interesting data to shed some light on the mechanisms that are responsible for the chromatin changes. For example, the analysis would greatly benefit from transcription factor (TF) motif analysis on promoters of genes with differential chromatin accessibility. Another option would be a digital genomic footprinting analysis on the gene panel of genes classified to be reversible on the niche, regardless of ATAC peak status. Such an analysis might identify TFs that play a role in the process of niche dependent signaling.

Minor

1. Regarding Figure 1c, indicating cell type proportions, changes are clearly visible. Please indicate significance of differences.

2. In line 224-227, the authors present the integrated analysis of the RNA-seq identified genes and the ATAC-seq data. How was the link between cCREs and genes made? Distance from TSS? How were multiple cCREs per gene handled?

3. SC clusters 1 and 2 presented in fig 2 b/g indicate a non uniform density for young and aged cells (left to right). Is this effect observed in non batch corrected cells as well?

4. When comparing marker expressions on fig 2 f/h, one can hardly see gene expression levels for cell groups due to the density lines plotted on the umap. In order to increase visibility, the authors might erase these lines, or change to a more comprehensive type of visualization, such as heatmaps.

5. Methods Genset Enrichment Analysis, line 712/713: I assume it should be downregulated/upregulated for (1) and (2)

Point-by-point response to reviewers

We would like to thank the reviewers for their remarks and their constructive comments and suggestions on our manuscript. We have now thoroughly addressed all comments of the four reviewers. We have also addressed their main concerns experimentally by providing 6 new figure panels, and 5 completely new supplemental figures (Fig. 1D, Fig. 4E, Fig. 5A-B, Fig. 6I-J, Supplemental Fig. 2, Supplemental Fig. 4, Supplemental Fig. 6, Supplemental Fig. 8, Supplemental Fig. 10, Supplemental Fig. 13). We believe that our addition of 12 additional transplantation RNA-Seq libraries to increase sample size greatly strengthens our main message which is that the niche is a critical regulator of the MuSC transcriptome. We have also edited the manuscript extensively. We believe, the revised version of our manuscript addresses all the concerns raised by the reviewers.

Reviewer #1 (Remarks to the Author):

The article by Lazure et al is an interesting manuscript that describes profiling of muscle stem cells (MuSCs) in several different conditions (at different ages and before and after transplantation into immunocompromised mice). The authors first assess changes in the transcriptomes of single MuSCs using single-cell RNA sequencing after FACS isolation. Next the authors isolate MuSCs from young and aged mice and perform transplantation into immunocompromised hosts. To evaluate changes in the transcriptome as a result of transplantation, muscle stem cells were isolated and profiled again by gene expression profiling. Last, the authors performed chromatin accessibility measurements and DNA methylation measurements using whole-genome bisulfite sequencing. Analysis of the datasets was performed, and the authors conclude there are variations in the epigenome with aging. Overall, the results are interesting but many of the generated datasets already exist dampening enthusiasm for value added to the field. Additionally, there is little to no validation of the datasets, several experiments are statistically under-powered, and many interpretations would need to be significantly scaled back. The primary limitation of the manuscript is that there are no validation experiments performed, and as written, the manuscript seems disjointed and a collection of datasets rather than a linear story. Major concerns are listed below for each of the figures and sections.

We thank the reviewer for these comments. While some of the datasets may already exist, such as scRNA-Seq of MuSCs, the novelty of this manuscript lies in the transplantation experiments. To our knowledge, MuSCs have never been transcriptionally profiled after re-isolation from an allogeneic host. Our additional replicates solidify our original conclusions that the niche is a principal regulator of the MuSC transcriptome. However, as per the reviewer's recommendation, we have scaled back our interpretations of some of our epigenomics data.

Figure 1a-1d: The authors use FACS to isolate MuSCs, fibro-adipogenic progenitors and macrophages and claim changes in the number of isolated cells. n=3 and n=4 replicates may be acceptable to perform statistical analysis of this population numbers, but this was not performed.

We ran several Dirichlet Regression models on the proportion data. The best fitting model showed a statistically significant shift in the proportions of the 3 cell types in aging ($p < 0.001$). The magnitude of the shift in the proportion of the 3 cell types due to aging can be visualized on the simplex plane (Figure 1D). In particular, the figure shows an age-related decrease in the proportions of MuSCs that is associated with an age-related increase in the proportion of FAPs. Details about the Dirichlet regression

analysis can be found here.

(https://csglab.github.io/transcriptional_reprogramming_muscle_cells/assets/notebooks/manuscript/di_richlet_regression.nb.html).

Additionally, FACS can be highly variable for enumerating cellular fractions and the authors should validate their claim of decreases in cell numbers through in situ staining.

We believe that validating the age-related decrease in MuSCs is unnecessary and redundant, since this decline in numbers has been previously characterized in multiple studies (Brack et al., 2005; Gibson and Schultz, 1983; Sajko et al., 2004; Sousa-Victor et al., 2014). Furthermore, the goal of our single-cell RNA-Seq data was not to confirm the decline in MuSC numbers with age, but rather to describe how subpopulations in a heterogeneous stem cell pool change with aging.

Moreover, given the claims in the manuscript about potential cellular crosstalk, the authors could quantify cellular distances between these cell types to determine if they are physically close and distances are altered with age.

The crosstalk between MuSCs and niche cells like macrophages and FAPs (Biferali et al., 2019; Mackey et al., 2017; Molina et al., 2021; Ratnayake et al., 2021), as well as the structure of the MuSC niche (Molina et al., 2021; Yin et al., 2013), have been widely studied and are beyond the scope of this paper. For example, Molina et al 2021 (Figure 2) already show FAPs and MuSCs in close proximity to one another through Pax7 and PDGFR α immunostaining. Similarly, Ratnayake et al 2021 show direct contact between MuSCs and macrophages post muscle injury.

Figure 1e-1j: The authors appear to have a single replicate for scRNA-Seq pooled from 3 young mice and 3 aged mice. This may be acceptable but there is no quality control presented for the single cell datasets (genes detected, UMIs detected, etc). As shown, it is unclear whether the resulting integrated nearest neighbor graph is accurate and there are three separate cell states or whether the authors' downstream analyses have produced this type of result.

The figure represents scRNA-seq data from 3 young and 4 old mice. We have individually performed the cell isolation, library preparation and scRNA-Seq for each of the 7 samples. We have now added an updated link containing all reproducible notebooks used for our scRNA-Seq analysis, including initial

Lazure et al. Point-by-point response to reviewers

quality control steps: https://csglab.github.io/transcriptional_reprogramming_muscle_cells/

Briefly, quality control was performed by filtering out cells that had a high proportion of reads coming from a small number of genes, a high percentage of mitochondrial reads, low total number of reads, outlier proportions of unspliced reads, or a high frequency of doublets. All subsequent steps of the analysis can be found on the website above as reproducible R notebooks.

As for the number of clusters, it is impossible to claim that a fixed number of clusters exist, since clusters can always be further subdivided. However, we settled on this clustering method based on biological relevance given the presence of an activated cluster, a pro-regenerative cluster, and a cluster upregulating stress response genes.

What is the nearest neighbor classification accuracy for these cell states if the authors used different integration strategies and hyperparameter choices? Given the availability of single cell datasets from mouse muscle stem cells at different ages, the authors could consider integrating their datasets with others to determine if population differences the authors are observing are also present. Inspection of other datasets such as from Tabula Muris Senis for example does not show segregation into 3 clusters. Are the differences observed and described here explained by differences as a result of experimental preparation?

Muscle stem cells are very rare, and comprise only a small fraction of the total cells within muscle tissue. As such, the Tabula Muris data contains 540 muscle satellite cells using their FACS isolation method, and 354 muscle satellite cells using their droplet method (tabula-muris.ds.czbiohub.org)

Contrarily, we used FACS to enrich for MuSCs in order to increase the number of stem cells. Our enriched data involves n=3 young mice containing 1261, 1936 and 1063 MuSCs, and n=4 aged mice containing 545, 692, 337 and 761 MuSCs for a total of 6595 scRNA-seq of MuSCs. Due to the significant increase in numbers of MuSCs available for clustering in our dataset, we have an increased capacity to resolve different clusters of MuSCs compared to other datasets by computational means.

Furthermore, we have now performed an additional quality control step by comparing the gene expression of our MuSCs, compared to the Tabula Muris MuSCs as a positive control, and Tabula Muris macrophages as a negative control (Supplemental Figure 2) . As expected, our MuSC dataset most closely resembles the Tabula Muris droplet MuSCs ($r=0.85$), and least resembles the Tabula Muris FACS macrophages ($r=0.52$).

Figure 2: The authors could experimentally demonstrate what physiological differences there are between the 3 groups of MuSCs (proliferative differences, engraftment efficiencies, surface markers or transcription factors expressed, etc). As presented, it is speculative if the observed differences are meaningful or an artifact of the bioinformatics processing. Additionally, many of the genes discussed have already been profiled in MuSCs and FAPs diminishing novelty of this set of results. The statement that aging impinges on common gene networks and pathways is not experimentally demonstrated with the data presented. The authors could validate their findings with RNAScope or immunohistochemistry additionally to determine if this statement is true. Lastly, it's very difficult to see some of the differentially expressed genes in Figure 2.

We have now added additional panels of UMAP Plots in Supplemental Figure 1 for known markers of quiescent MuSCs. We also go into more detail in the manuscript on the differences between individual clusters. To increase readability, we have also split our previous figure 2 into two figures (new Figures 2 and 3).

In order to validate our RNA-Seq data, we have performed immunofluorescence staining on young and aged EDL-associated MuSCs for two cell cycle genes (Supplemental Figure 4). For example, our scRNA-Seq data displays significant heterogeneity in the expression of Ccnd1. Similarly, young MuSCs show heterogeneity in the level of expression of cyclin D1 protein, as shown by fluorescence intensity level.

Figure 3: The authors used a transplantation model to decouple the effects of signaling from the niche into MuSCs and performed gene expression profiling 21 days after transplantation. This strategy is interesting but given the tremendous amount of cellular death from transplantation, it seems as though the strategy selects for a subset of cells. Additionally, recent evidence (Liu et al, Cell Stem Cell 2018) has shown that aged activated MuSCs are more prone to cellular death than young activated MuSCs. Thus, the transplantation assay seems to “select” cells rather than assaying the true diversity that exists in aged animals. As an alternative, the authors could perhaps do validation of what state the MuSCs that remain are in (are they quiescent, still activated?). This would be helpful to argue if differences observed are because MuSCs are mostly still activated or have returned to quiescence at the time profiled.

To answer this question, we have performed immunofluorescence staining of host transplanted mouse TA muscles for Ki67 (Figure 4E). As shown in this new panel, Pax7⁺/GFP⁺ donor cells are Ki67-negative

compared to our positive control (cardiotoxin-injured TA muscle), where a number of Pax7⁺ MuSCs are Ki67⁺.

Furthermore, when counting the number of GFP⁺/PAX7⁺ donor MuSCs that are Ki67⁺ per cross-section, the number is most frequently 0. Additionally, using our RNA-Seq data of Young and Aged MuSCs before and after transplantation, we show that T21 cells are not expressing high levels of transcripts for Ki67, or Myogenin, another marker of activated/differentiating MuSCs.

The authors could also consider increasing their sample size as *n*=2 seems low given the variance plotted in their PCA plots and Spearman correlations did not seem to be reported. The authors could also somehow try to connect their datasets to the single cell datasets (such as staining to determine if transplantation results in enhancement of interactions between cell types or decreases).

We agree with the reviewer that increasing sample size is valuable to this study. Therefore, we have performed 6 additional, independent MuSC transplantation and re-isolation experiments, and have generated and sequenced 12 new SMART RNA-Seq libraries (referred to as batch 2 in Figure 5 and Supplemental Figure 8). Considering both batches of samples, we now have *n*=6 for Young T21, *n*=5 for Young T21, *n*=5 for Aged T0 and *n*=5 for Aged T21.

We have generated pairwise correlation scatterplots to assess the similarity of the new batch with the first batch performed and saw a strong correlation between most samples. Sample aged_T0_05 and

aged_T21_05 were removed due to lower than normal correlation with other samples within the group, leaving us with n=5 Aged T0 and T21 samples.

Similarly, when we assess the correlation between age-altered genes in our single cell RNA-Seq data compared to our bulk RNA-Seq data, we see consistency in age-related genes from both datasets (e.g. *Ccl11*, *Grid 2*, *Dcn*, *Sprc*, *Ccmd1*, *Gas1*, *Col3a1*, among others) when we analyze batch 1 alone, or batch 1 and 2 together.

Lastly, a recent manuscript was published that (Evano et al PLoS Genet 2020) performed similar studies whereby MuSCs were profiled using RNA-Seq and DNA methylation before and after transplantation but the authors did not cite this paper.

As another paper employing transplantation, we have now cited this paper in the introduction. While similar in concept, Evano et al analyze the ability of extraocular muscle stem cells to engraft and be reprogrammed by the hindlimb environment. In other words, while Evano et al are varying the cell location of origin, we are varying the age of the host.

Figure 4: Similar to other experiments above, many of the identified variations from the experiment are not validated and the conclusion of the experiment, which is that “the niche is a principal regulator of the MuSC transcriptome,” and that “a significant portion of the transcriptome of MuSCs can be reprogrammed back to a youthful state by exposure to a young niche milieu” seems premature. If genes are reversibly changed, can the authors validate these results first at the protein level and then by rescue in aged animals? The authors don’t need validate many but 1-2 predicted targets seem reasonable. The authors could also use their single cell datasets to identify potential signaling cells that are altered and which may be deleterious and which may advantageous for MuSCs.

We agree that the distinction between changes in the transcriptome and protein level should be clarified. Throughout the manuscript, we mention that MuSCs can be transcriptionally reprogrammed, to make this distinction.

With regards to validating our transcriptome data at the protein level, there is a technical limitation with regards to the very low number of cells re-isolated after engraftment. We have re-isolated between 30 and 400 cells, with the median number of re-isolated cells being ~100. As such, serial transplantation or in vitro assays to assess functionality post transcriptional-reprogramming are technically impossible, without pooling multiple aged mice which are difficult to procure. However, we make it clear that the scope of this paper includes transcriptional changes, not changes at the protein or functional level.

Figure 5: The authors claim about differential RNA stability should be validated with an orthogonal assay for several transcripts through qPCR. If there are differences manifesting from deposition of cytosine methylation between young and aged MuSCs but the authors are performing the assay on myoblasts which have different methylation profiles, the authors should discuss this aspect in their results section. As written, this section seems underdeveloped and should be connected to enzymes that deposit remove or deposit these modifications. The enrichments of the ATAC-Seq assay look to be strong, but the authors may consider using the Irreproducibility Discovery Rate (IDR) framework to look at agreement between peak calls. The comment that genes that are niche-responsive are transcriptionally fluid seems like an over-statement and the data do not effectively reflect this claim. Given there are several ATAC-Seq (Garcia-Prat et al, Nature Cell Biol. 2020 & Shcherbina et al, Cell Rep 2020), and DNA methylation datasets on young and aged MuSCs (Hernando-Herraez et al, Nature Comm 2019, Evano et al PLoS Genet 2020), the authors could consider integrating these datasets and observe if the variations in their datasets are unique or similar.

For the DNA methylation analysis, we opted to use primary myoblasts instead off freshly-isolated MuSCs due to the requirement for high quantities of genomic DNA and the low number of MuSCs that can be isolated from aged mice. We now briefly clarify this point in the manuscript.

With regards to DNA methylation and aging, Hernando-Herraez et al. show an increase in cell-cell heterogeneity, and specific changes in DNA methylation, but no net change in levels of DNA methylation. Our data corroborates this, since we also show that there are age specific DMRs but no net gain or loss of methylation in aging (Figure 6E).

In light of additional data coming from our increased number of replicates, we have edited the text related to figure 6 to avoid making overstatements. We now simply claim that there is an association between chromatin accessibility and the specific category of upregulated ARIs.

Lazure et al. Point-by-point response to reviewers

Reviewer #2 (Remarks to the Author):

The manuscript by Lazure et al. addresses a significant question in stem cell biology i.e. whether the age-related decrease in stem cell function is due to 'intrinsic' changes or the 'extrinsic' cues from an aged niche. They utilize single cell RNA-seq data and analyses to dissect out the relative contributions of these 'intrinsic' and 'extrinsic' changes and come to the really interesting conclusions that the 'intrinsic' changes appear to be largely driven by epigenetic changes in the muscle stem cells.

The study is interesting and significant, however there are some concerns/suggestions as follows that need to be addressed and revised:

1. The authors have nicely delineated the different clusters of MuSC that alter with aging in abundance; it had not been described how the old MuSC clusters change in abundance or quality upon transplantation in the young niche. For example, is the young niche capable of reprogramming old MuSCs into the cluster 3 MuSCs that are lost with age? How do the abundance of MuSC 1 and 2 change in this experiment?

We thank the reviewer for their positive comments on our manuscript. It is difficult to assess how subcluster abundance changes in transplanted cells due to the bulk library preparation of the transplanted samples using SMART-Seq. Similarly, MuSC clusters 1 and 2 are quite similar with respect to expression profile of many genes, and only separate out due to our high number of stem cells and therefore the ensuing resolution after selectively enriching for MuSC by FACS. In other words, MuSC 1 and MuSC 2 are different based on their transcriptomic profile, but do not possess marker genes that are solely expressed in one cluster over the other.

As for the question about restoring cluster 3, this has now been answered in Supplemental Figure 10. We can observe a partial restoration of some, but not all markers of MuSC Cluster 3 after transplantation into a young niche. This is indicative that we may be restoring the expression of the cluster after transplantation, but since we do not have scRNA-Seq data of re-isolated cells, we cannot definitively confirm this observation.

2. It would be very informative to analyze separately how the MuSC 1 and 2 clusters respond to the young niche? Are there pathways that are specific to one vs another?

While MuSC1 and MuSC2 vary based on the level of expression of certain genes, they do not have unique markers whereby one cluster expresses a gene that has 0 expression in the other. Therefore, it is not possible to effectively determine if one of these clusters is increasing or decreasing after engraftment into the young niche.

3. The rationale for using myoblasts derived from MuSCs for the Methylation analyses is not clarified-it would be much more relevant to directly compare the epigenetic patterns and gene expression data in the same cell types i.e. MuSCs in which the RNA-seq and ATAC seq are done. This is an important point since it affects the central conclusion of the study.

Lazure et al. Point-by-point response to reviewers

While we agree with the reviewer that the methylation analysis would be more relevant on freshly-sorted MuSCs, we opted to use myoblasts due to the technical difficulty of sorting enough MuSCs to obtain sufficient genomic DNA (2 µg of DNA as required for WGBS). We therefore needed to expand myoblasts to P3 in culture to obtain enough genomic DNA for bisulfite conversion and library preparation/sequencing.

However, DNA methylation is a relatively stable epigenetic mark, and myoblasts express ample amount of DNMT1 (Liu et al., 2016), which is responsible for the propagation and maintenance of DNA methylation. Therefore, we believe that using myoblasts is sufficient in this scenario.

Reviewer #3 (Remarks to the Author):

In this manuscript, the authors compared gene expression profiles between young and old muscle stem cells (MuSC) via single-cell RNA-seq (scRNA-seq) and demonstrated that there are three clusters (clusters 1, 2, and 3). Cluster 1 (MuSC1) is more prone to age-related loss, whereas the cells in Cluster 2 (MuSC2) display enhanced retention with age. Cluster 3 (MuSC3), which is characterized by genes involved in MuSC activation such as MyoD1 and Cdkn1c is completely lost in aged mice. MuSC transplantation of young and aged cells into heterochronic host muscles demonstrated that the expression of half of the age-altered genes could be restored by exposure to a young niche environment. Unaltered genes exhibit altered chromatin accessibility shown by ATAC-seq and Methylome. By contrast, the majority of age-related altered genes are not epigenetically encoded and restorable by exposure to a young niche environment. Therefore, the authors concluded that the stem cell niche might restore the age-related gene expression dependent on chromatin accessibility.

This is an excellent and unique approach for age-related gene expression and alteration by the age-associated niche environment. Before publication, the authors should address the following issues.

1. While MUSC3 is mentioned as an activated MuSC population, the characteristics of MUSC1 and MUSC2 are not so clear. Would you please explain which kind of clusters they are and which representative genes with both clusters should be indicated?

While the MuSC1 and MuSC2 clusters are transcriptionally very similar, we now go into more details in the manuscript to describe the differences present, focusing on the FOS/JUN/EGR2 pathways, antioxidant/stress response genes, and the OSMR/STAT3 pathways (Figure 2, supplemental figure 5). Overall, we characterize, Cluster 1 as pro-regenerative cells, Cluster 2 as stress-responsive cells, and Cluster 3 as activated cells. This updated characterization is elaborated upon in the revised text line 122-149.

2. In Supplementary Table 2, sample numbers of scRNA-seq for MuSC transplantation of young and aged cells into heterochronic host muscles include Aged T0 (x2), Aged T21 (x3), Young T0 (x3), and Young T21 (x2). This reviewer feels triplicated sample number is required for each group since some gene expressions, especially genes for key myogenic factors and aging-associated genes, display significant variability, maybe due to the sample preparations or timing. For example, Myog: (Aged T0: 2.0275, 4.5914), (Aged T21: 0.036562, 39.6274), (Young T0: 1.0838, 20.6157, 194.1455), (Young T21: 25.5482, 72.7847).

Lazure et al. Point-by-point response to reviewers

We agree that increasing sample number is necessary for this paper. We have now increased the number of biological replicates by performing additional transplantation experiments (referred to as batch 2 in Figure 5). We now have n=5 Aged T0, n=5 Aged T21, n=6 Young T0 and n=5 Young T21 biological replicates. We believe these additional replicates solidify our conclusions that the niche is the principal regulator of MuSC gene expression. Furthermore, factors in the young niche have the ability to restore the transcriptome of transplanted aged MuSCs.

3. Extended Data Figure 4e shows 3x Young T21 samples, but in Supplementary Table 2 and Extended Data Figure 4f shows 2x Young T21 samples. Would you please explain this discrepancy?

One of the Young T21 samples has been removed due to its status as an outlier after careful analysis, therefore we only kept n=2 for the initial analysis. We have since sequenced additional samples to now have n=5 Young T21 samples. New figures have been recreated to incorporate the additional replicates (Figure 5A,B,D,E, Supplemental Figure 8).

Reviewer #4 (Remarks to the Author):

The manuscript from Lazure et al investigates the transcriptional and epigenetic landscape of muscle stem cells. The work focuses on the process of stem cell aging, and within this context on the role of the stem cell niche that is well known to play a crucial role in stem cell maintenance and cell quiescence. By using an elegant experimental design and sorting strategy, as well as screens on the transcriptome, chromatin accessibility, and DNA methylation level, Lazure and colleagues were able to unravel differentially regulated entities respectively and to classify the given changes into age dependent, and stem cell niche dependent changes. Finally, a transplantation approach with young and aged stem cells delivered into radiated young mice allowed for the investigation of reversible gene expression patterns in dependency of the young niche. Of note, Lazure and colleagues found distinct subtypes of stem cells and FAPs by SC-RNA-seq and a unique subpopulation of MuSC exclusively given in young mice. Further, they were able to define a reversible and irreversible gene panel in dependency of a young niche. Finally, they found evidence for epigenetically coded (DNA methylation, accessibility) age-deregulated alterations to be less responsive to the influence of the niche.

Comments to the authors

Major

1. The authors use cell sorting followed by 10x based SC-RNA-seq to investigate cell subtypes and subtype proportions for MuSC, FABs and Macrophages. In case of MuSC, three and two cell clusters for young and aged MuSC are defined respectively (fig1e, fig2b/g). This is for sure a very interesting finding, and the reader is eager to learn about the potential role of these subtypes. The missing cluster three in old MuSC is of course of particular interest. Unfortunately, a more comprehensive characterization of these clusters according to widely accepted concepts of MuSC regulatory circuits as e.g. described in Tierney/Sacco 2016 (<https://doi.org/10.1016/j.tcb.2016.02.004>) is critically missing. In particular, a closer look on MuSC cluster 3, missing in aged mice, and its potential role in MuSC self-renewal would definitely strengthen this manuscript. For example, Myf5 is often used to define a long term self renewing MuSC population, and its expression is shown in ext fig1 i. Can one state that MuSC cluster 3 expresses low levels of Myf5 compared to Cluster 1 and 2 ? Along this line, fig2 presents Cdkn1c as marker for cluster 3, known to be activated in induced MuSC starting to become myoblasts. As it stands

now, the presented data is very interesting, but it opens up the room for a multitude of obvious questions that are not answered or discussed.

MuSC Cluster 3 does indeed express lower levels of Myf5 compared to Clusters 1 and 2. MuSC Cluster 3 also expressed markers of activation/differentiation such as elevated MyoD1 and Myogenin (Supplemental Figure 1). As described in Tierney/Sacco 2016, Myogenin protein expression leads to the advancement into the myocyte stage of differentiation. Additionally, Cluster 3 expresses reduced expression of other known quiescence markers such as Sdc4, Calcr, Chrdl2, and Notch3. Therefore, the lower levels of Myf5 in cluster 3 does not appear to be associated with self-renewing, non-differentiating cells. Rather, cluster 3 are committed towards the entry into the differentiation pathway. In the revised manuscript, we more deeply characterize the different MuSC clusters, as can be seen in lines 122-149.

2. The SMART RNA-seq from engrafted aged MuSC is compared and characterized in fig4 /ext4/ext5, and the author indicate a panel of reversible genes that probably push back the aged MuSC to a more young phenotype. Of course, it would be interesting to know if these genes (or some of them) overlap with the “missing” cluster 3 of MuSC as shown in fig2b.

In Extended Data Figure 10, we now show that some, but not all markers of MuSC Cluster 3 are partially restored in aged MuSCs after engraftment into young mice. This may be somewhat indicative of the restoration of cluster 3. However, other markers of cluster 3 are not restored after engraftment. Also, given the small number of cells within Cluster 3, it is difficult to conclude whether or not gene reversibility post-engraftment is due to MuSC Cluster 3 restoration.

It is clear that a SC approach is not possible due to low cell numbers from engrafted animals, however, a bioinformatics approach for bulk RNA deconvolution such as given with cybersortx (Newman Lab Stanford) might help to answer this question.

We attempted to deconvolute the bulk RNA-Seq using MusiQ (X. Wang, Nature Comm 2019). However, this yielded unclear results, most probably due to the fact that our 3 clusters of MuSCs rarely have

distinct, unique marker genes; rather, they vary in the level of expression of various genes. Therefore, deconvolution of bulk RNA-seq onto scRNA-seq in this case yielded inconclusive results.

3. Figure 5 deals with genome wide DNA methylation data and chromatin accessibility based on ATAC-seq. On line 206-208, the authors state that they found a significant over-representation of irreversible upregulated genes among loss-of-methylation DMRs, and I assume that Figure 5e and ext. table 2 show the same data. It is illustrated that all combinations of up/down reversible/irreversible contain approximately the same average number of loss and gain DMRs. However, the complete analysis is based on a small number of overlapping genes, namely 15 and 17 respectively (ext. table 2). In this context it is not clear what is shown exactly. For example, looking at the down/reversible bar (second from left), one can observe gain and loss at about 50% from the total height of the bar, that reflects ~0.9 DMRs per gene, but in table ext 2 this column indicates 0 and 1 gene.

In the extended data table, we are looking at how many genes are associated with the list of age-related DMRs (therefore, we can only have 1 gene per DMR). In figure 6, we are looking at the average number of DMRs per gene (therefore, we can have multiple DMRs associated with a single gene).

4. In figure 5i, the authors show two sets of ATAC-seq peaks - increased and decreased accessibility by age, in total ~7560 peaks. On the right side, the overlap with the up- and down-regulated genes is shown, as well as an enrichment of the irreversible set of genes in both sets of peaks. However, the analysis assumes that gain-of-accessibility only leads to upregulation of genes. For completeness, it would be great if the authors would also show the overlap of loss-of-accessibility with up-regulated genes and gain-of-accessibility with down-regulated genes. I assume that the numbers in principle should be given in ext. table 3, but I was not able to find the gene numbers given in the pie chart.

For completeness, we have now added Figure 6J, to see how many differentially accessible peaks (including both increased and decreased accessibility) within various distances from the TSS, are associated with reversible and irreversible genes. We see that, as we move closer to the TSS (within 10kb), there is an increasing enrichment of accessible peaks associated with age-upregulated irreversible genes. A similar, yet less striking pattern is observed with age-downregulated irreversible genes and peaks with decreased accessibility. In any case, we have edited the manuscript to avoid making overstatements and instead describe the associations we observe.

5. Having a closer look on ext table 3, one also get the impression that for upregulated peaks not only irreversible genes are significant, rather than all upregulated genes.

In case of the downregulated peaks and downregulated genes the reversible genes seems to be more significant than the irreversible, that do not fit to fig5i at all. Please clarify.

In light of new data arising from increasing our number of replicates (we have sequenced 12 additional libraries for the transplantation experiments), we now see that overall, downregulated genes are enriched within peaks of decreased accessibility, and upregulated genes are enriched within peaks of increase accessibility in aging. As we move closer towards the TSS (within 10kb), we begin to notice a pattern whereby upregulated irreversible genes become more strongly associated with upregulated peaks.

6. Along this line, ext. fig. 6e shows the majority of ATAC peaks to be located in proximity to genes (promoter <1kb, introns, exons, UTR). In addition, ext. fig. 6a indicates ~50% of all TSS regions (and probably promoter/gene bodies, scale not indicated) to be accessible. Having these numbers, I'm wondering why the number of genes up/downregulated overlapping a differential peak is "just" in the range of ≤ 100 (fig 5i right) each. Does this mean the majority of differentially regulated genes is in closed chromatin, or in not changing chromatin regarding accessibility? If this is the case, can one hold generalized terms such as "that changes in chromatin accessibility and DNA methylation impede transcriptional restoration by niche exposure"?

This is a very good point that reviewer is raising. However, not all the changes we observed in the transcriptome are associated with significant changes in chromatin accessibility at various thresholds. These changes may therefore be due to more transient signaling mechanisms. Although we think that changes in chromatin accessibility might impede transcriptional restoration as pointed out by the reviewer, in the absence of 3D genome data we cannot make a conclusive statement on this important point. This is because most of the alteration in chromatin and DMRs map to distal regulatory regions. In the absence of 3D data which can show causal relationship between distal enhancers and their target TSS, one can only make association based on proximity. As such, the conclusion can at best be correlative and need formal experimental proof.

7. Fig 5j gives a scheme on chromatin accessibility and genome responsiveness to the niche, in dependency on up/down regulation of genes. As it is given now, the scheme indicates that down regulated genes per se have a higher degree of chromatin accessibility. Further one can conclude that all upregulated genes are more responsive to a niche than downregulated genes. Is this intended by the authors? I recommend to split the figure or find a different way of visualization.

To avoid confusion, we have removed the schematic diagram.

8. While it might not be possible to perform additional epigenetic screens in the engraftment-model (also due to the low amount of recovered MuSC), it would be beneficial if the authors would at least discuss the limitations of comparing natural epigenetic markers of aging with the change in niche environment used for the RNA-seq experiments.

We agree with the reviewer about this limitation on our conclusions derived from our epigenetic screens. The ideal experiments would have been to assess the epigenetic state (DNA methylation and ATAC-Seq) of aged MuSCs before and after engraftment into the young niche. However, due to the low number of cells recovered, this would be impossible, hence our comparison of T0 non-engrafted young and aged MuSCs. We have therefore edited the manuscript to explain the strength and the limitations of our data.

9. Figure 5h illustrates ATAC signals at DMR positions surrounded by a 10kb window, with one panel indicating gain of methylation spots, and a second panel indicating loss of methylation spots (left panel). It is not clear to me what can be concluded from this visualization. One can speculate that the overall chromatin accessibility change in young cells is smaller than in aged cells? Why did the author choose a 10KB window, I would expect this size to include further CpGs. Did the authors follow an approach that combines individually changed CpGs to annotated regions such as promoters or tiles in order to find larger consistently changed DNA regions regarding methylation?

In that figure, we are simply showing the link between age-related DMRs and changes in chromatin. The figure shows that loss of methylation on age-related DMRs are associated with an opening of chromatin when overlaid with ATAC-seq peaks.

10. Regarding the ATACseq data analysis as it stands, I feel the authors miss to use this very interesting data to shed some light on the mechanisms that are responsible for the chromatin changes. For example, the analysis would greatly benefit from transcription factor (TF) motif analysis on promoters of genes with differential chromatin accessibility. Another option would be a digital genomic footprinting analysis on the gene panel of genes classified to be reversible on the niche, regardless of ATAC peak status. Such an analysis might identify TFs that play a role in the process of niche dependent signaling.

We have now performed ATAC-Seq foot printing using TOBIAS (Bentsen et al., 2020), which we have added to Extended Data Figure 12. The interactive figure shows TF consensus motifs in ATAC-seq peaks that are differentially enriched between young up (red) and aged up (green).

Minor

1. Regarding Figure 1c, indicating cell type proportions, changes are clearly visible. Please indicate significance of differences.

We have now included a Dirichlet Regression model to better illustrate the changes in cell type proportions in aging (Figure 1D)

2. In line 224-227, the authors present the integrated analysis of the RNA-seq identified genes and the ATAC-seq data. How was the link between cCREs and genes made? Distance from TSS? How were multiple cCREs per gene handled?

Each differentially accessible peak (cCRE) was assigned one gene using the GREAT software, which is based on distance from the TSS. If more than one peak is assigned to the same gene, that gene is only counted once in our analysis. We have updated the Method section (line 490-502, subsection: Analysis of differentially accessible ATAC-seq peaks) to provide more details.

3. SC clusters 1 and 2 presented in fig 2 b/g indicate a non-uniform density for young and aged cells (left to right). Is this effect observed in non-batch corrected cells as well?

Without batch correction, the young and aged cells would be clustered further apart since the aged and young samples belong to different batches. The batch correction tries to bring the cells from the two conditions into a common mutual nearest neighbors (MNN) space. In the revised manuscript we have described how the batch correction was performed (line 537-550).

4. When comparing marker expressions on fig2 f/h, one can hardly see gene expression levels for cell groups due to the density lines plotted on the umap. In order to increase visibility, the authors might erase these lines, or change to a more comprehensive type of visualization, such as heatmaps.

We have split this figure into 2 separate figures (one figure for cluster-related changes and one figure for age-related changes) in order to increase size and visibility.

5. Methods Genset Enrichment Analysis, line 712/713: I assume it should be downregulated/upregulated for (1) and (2)

This typo has been corrected in the methods section.

Bentsen, M., Goymann, P., Schultheis, H., Klee, K., Petrova, A., Wiegandt, R., Fust, A., Preussner, J., Kuenne, C., Braun, T., *et al.* (2020). ATAC-seq footprinting unravels kinetics of transcription factor binding during zygotic genome activation. *Nature Communications* *11*, 4267.

Biferali, B., Proietti, D., Mozzetta, C., and Madaro, L. (2019). Fibro-Adipogenic Progenitors Cross-Talk in Skeletal Muscle: The Social Network. *Front Physiol* *10*, 1074.

Brack, A.S., Bildsoe, H., and Hughes, S.M. (2005). Evidence that satellite cell decrement contributes to preferential decline in nuclear number from large fibres during murine age-related muscle atrophy. *Journal of Cell Science* *118*, 4813-4821.

Gibson, M.C., and Schultz, E. (1983). Age-related differences in absolute numbers of skeletal muscle satellite cells. *Muscle & nerve* *6*, 574-580.

Liu, R., Kim, K.-Y., Jung, Y.-W., and Park, I.-H. (2016). Dnmt1 regulates the myogenic lineage specification of muscle stem cells. *Scientific Reports* *6*, 35355.

Mackey, A.L., Magnan, M., Chazaud, B., and Kjaer, M. (2017). Human skeletal muscle fibroblasts stimulate in vitro myogenesis and in vivo muscle regeneration. *The Journal of physiology* *595*, 5115-5127.

Molina, T., Fabre, P., and Dumont, N.A. (2021). Fibro-adipogenic progenitors in skeletal muscle homeostasis, regeneration and diseases. *Open Biology* *11*, 210110.

Ratnayake, D., Nguyen, P.D., Rossello, F.J., Wimmer, V.C., Tan, J.L., Galvis, L.A., Julier, Z., Wood, A.J., Boudier, T., Isiaku, A.I., *et al.* (2021). Macrophages provide a transient muscle stem cell niche via NAMPT secretion. *Nature* *591*, 281-287.

Sajko, S., Kubínová, L., Cvetko, E., Kreft, M., Wernig, A., and Erzen, I. (2004). Frequency of M-cadherin-stained satellite cells declines in human muscles during aging. *The journal of histochemistry and cytochemistry : official journal of the Histochemistry Society* *52*, 179-185.

Sousa-Victor, P., Gutarra, S., García-Prat, L., Rodríguez-Ubreva, J., Ortet, L., Ruiz-Bonilla, V., Jardí, M., Ballestar, E., González, S., Serrano, A.L., *et al.* (2014). Geriatric muscle stem cells switch reversible quiescence into senescence. *Nature* *506*, 316-321.

Yin, H., Price, F., and Rudnicki, M.A. (2013). Satellite cells and the muscle stem cell niche. *Physiological reviews* *93*, 23-67.

Reviewer #1 (Remarks to the Author):

The article by Lazure et al is an interesting manuscript that describes profiling of muscle stem cells from young and aged mice before and after transplantation into immunocompromised mice. Unfortunately, many claims in the manuscript remain overstated, much of the data and observations have been observed in other published manuscripts and a critical limitation that was previously brought up was not addressed. Specifically, the transplantation experiments themselves select for cells that are less prone to cell death, which has been shown to increase with aging. This limitation is demonstrated in the section on transplantation whereby 10-20k cells are implanted and only 30-400 are recovered. Even if the limitations with the proposed experiment are rejected, there is still no validation of genes found, how they may influence muscle stem cell states and regenerative properties that change with age. The connection to changes in chromatin also remain somewhat weak and could be improved. Given the authors claim their major contribution to the field is the datasets produced after transplantation, publication of the manuscript at this stage would be premature.

The techniques used in the manuscript are well established and have been used extensively for the last 5-10 years. The significance of the findings is not strong given many genes profiled have already been found in single cell RNA sequencing datasets of muscle stem cells. The DNA methylation and ATAC-Seq experiments have already been performed on young and aged muscle stem cells making the contribution to the literature the new bulk RNA-Seq datasets after transplantation. The manuscript still seems disjointed and a collection of data rather than a story, still contains several overstatements such as "we found age-related loss of DNA methylation is associated with loss of heterochromatin," and "we found niche-irresponsive genes." The authors do not measure any histone modifications associated with heterochromatin, but rather found changes in accessibility with ATAC-Seq. If a gene is not sensitive to the niche, the authors should knock the gene out globally and determine no change in the muscle stem cell niche occurs as well as the muscle stem cell themselves. Lastly, there is still minimal validation of the datasets.

The references have been updated, but conclusions seem overstated. Why not follow up on a gene or two and really characterize their role in the niche or use the FAPs and macrophage data generated from Figure 1 (which is not really discussed much) and identify what ligand may be responsible for altering changes in MuSCs with aging?

Reviewer #2 (Remarks to the Author):

The authors have addressed all critiques adequately.

Reviewer #4 (Remarks to the Author):

The authors answered and addressed all my questions and the manuscript was substantially improved by the additional analysis and edits that were introduced. I feel all my requests to be considered in the manuscript now and suggest the editor to publish the manuscript in NC

Reviewer #1 (Remarks to the Author):

The article by Lazure et al is an interesting manuscript that describes profiling of muscle stem cells from young and aged mice before and after transplantation into immunocompromised mice. Unfortunately, many claims in the manuscript remain overstated, much of the data and observations have been observed in other published manuscripts and a critical limitation that was previously brought up was not addressed.

Response: We thank the reviewer for their appraisal and for their critical review of our manuscript. We are aware that multiple studies have performed ATAC-seq and scRNA-seq on muscle stem cells. However, the focus of our manuscript is direct quantification of the effect of MuSC's niche environment on gene expression. We have carefully designed and incorporated allogeneic stem cell transplantation coupled with Switching Mechanism at 5' End of RNA Template to identify niche- reversible and irreversible age-altered genes. This is the first study to do that.

Specifically, the transplantation experiments themselves select for cells that are less prone to cell death, which has been shown to increase with aging. This limitation is demonstrated in the section on transplantation whereby 10-20k cells are implanted and only 30-400 are recovered.

Response: We now provide a new supplemental figure 7 which shows that there is no statistically significant effect of age of the donor cells on their engraftment by the analysis of 5 biological replicates per groups in young and aged. This data is also consistent with previously published data by an independent group (Novak et al. 2021, Aging Cell, Human muscle stem cells are refractory to aging) which shows that muscle stem cells from xenografts of various ages have equal engraftment potential. The low engraftment efficiency that the reviewer is pointing out is well known in the field and is independent of the age of the donor cells as shown in supplemental figure 7.

Supplemental Figure 7

a

Biological Replicate	# of MuSCs Transplanted	# of MuSCs Reisolated	Percentage (%) Reisolated
Young 1	20,000	75	0.375
Young 2	20,000	20	0.10
Young 3	15,000	150	1.00
Young 4	20,000	73	0.365
Young 5	20,000	104	0.520
Aged 1	20,000	44	0.220
Aged 2	20,000	66	0.330
Aged 3	10,000	101	1.01
Aged 4	20,000	33	0.165
Aged 5	20,000	35	0.175

Supplementary Fig. 7: Age effect on donor MuSCs transplantation. *a*, number of engrafted donor cells from young and aged in five biological replicates. *b*, two-tailed *t*-test showing no statistically significant age effect of donor cells on engraftment efficiency.

Even if the limitations with the proposed experiment are rejected, there is still no validation of genes found, how they may influence muscle stem cell states and regenerative properties that

change with age. The connection to changes in chromatin also remain somewhat weak and could be improved. Given the authors claim their major contribution to the field is the datasets produced after transplantation, publication of the manuscript at this stage would be premature.

Response: The focus of our manuscript is to quantify the effect of niche environment on muscle stem cell gene expression. Although it is important to study how restoration of age-altered genes affects the regenerative function of muscle stem cells, it is beyond the scope of this manuscript.

The techniques used in the manuscript are well established and have been used extensively for the last 5-10 years. The significance of the findings is not strong given many genes profiled have already been found in single cell RNA sequencing datasets of muscle stem cells. The DNA methylation and ATAC-Seq experiments have already been performed on young and aged muscle stem cells making the contribution to the literature the new bulk RNA-Seq datasets after transplantation.

Response: We respect the opinion of the reviewer. However, this manuscript squarely focuses on the contribution of the niche environment to MuSC gene expression. The data and conclusions which states that the niche environment is a key regulator of MuSC gene expression and that a significant number of age-altered genes are restorable by exposure to young niche are very compelling. We provide the additional data such as ATAC-seq, WGBS and scRNA-seq to strengthen the manuscript.

The manuscript still seems disjointed and a collection of data rather than a story, still contains several overstatements such as “we found age-related loss of DNA methylation is associated with loss of heterochromatin,” and “we found niche-irresponsive genes.” The authors do not measure any histone modifications associated with heterochromatin, but rather found changes in accessibility with ATAC-Seq.

Response: The manuscript has been read and edited by all authors and has gone through internal review at the Lady Davis Institute for clarity and consistency. We have done further edited in this revised version to address the reviewer’s concern and to meet the editorial guidelines. We respectfully disagree with the reviewer stating that we “overstate” that loss of DNA methylation is associated with loss of heterochromatin. Our statement is fully supported by Figure 6f-g. Regarding reviewer’s comment on the quantification of histone modifications, this request goes beyond the scope of the current manuscript.

If a gene is not sensitive to the niche, the authors should knock the gene out globally and determine no change in the muscle stem cell niche occurs as well as the muscle stem cell themselves. Lastly, there is still minimal validation of the datasets.

Response: Although generating knockout mice from candidate genes and characterizing their effect on muscle stem cell function is important for future studies, at this stage it is extraordinarily unrealistic to be included and goes far beyond the scope of the manuscript.

Lazure et al. Point by point response to reviewers

The references have been updated, but conclusions seem overstated. Why not follow up on a gene or two and really characterize their role in the niche or use the FAPs and macrophage data generated from Figure 1 (which is not really discussed much) and identify what ligand may be responsible for altering changes in MuSCs with aging?

Response: We thank the reviewer for suggesting the ligand/receptor interaction analysis. We now provide a supplemental figure 6 in which we used our scRNA-seq data from MuSCs, FAPs and macrophages for cell-cell interactions. This new analysis shows that receiver/emitter interactions involving ECM decline in aged cells, while chemokine signaling is increased in aged cells. Moreover, it shows that FAP-FAP, MuSC-FAP, and FAP-MuSC cell interactions are over-represented in young samples. We also see that MuSCs and Macrophages are over-represented as emitters for the young samples. The MuSC-MuSC and Macrophage-Macrophage interactions are over-represented in aged samples.

Supplemental figure 6

Reviewer #2 (Remarks to the Author):

The authors have addressed all critiques adequately.

Response: Thank you very much!

Reviewer #4 (Remarks to the Author):

Lazure et al. Point by point response to reviewers

The authors answered and addressed all my questions, and the manuscript was substantially improved by the additional analysis and edits that were introduced. I feel all my requests to be considered in the manuscript now and suggest the editor to publish the manuscript in NC

Response: Thank you very much!

Reviewer #2 (Remarks to the Author):

Authors have satisfactorily addressed the concerns of all reviewers.

Lazure et al. Point by point response to reviewers

Reviewer #2 (Remarks to the Author):

Authors have satisfactorily addressed the concerns of all reviewers.

Response: thank you very much!